# Uncovering the cytochrome P450-catalyzed methylenedioxy bridge formation in streptovaricins biosynthesis

Guo Sun [1], Chaoqun Hu[1], Qing Mei[1], Minghe Luo[1], Xu Chen[1], Zhengyuan Li[1], Yuanzhen Liu[1], Zixin Deng[1], Zhengyu Zhang [1✉] & Yuhui Sun [1✉]

Streptovaricin C is a naphthalenic ansamycin antibiotic structurally similar to rifamycins with potential anti-MRSA bioactivities. However, the formation mechanism of the most fascinating and bioactivity-related methylenedioxy bridge (MDB) moiety in streptovaricins is unclear. Based on genetic and biochemical evidences, we herein clarify that the P450 enzyme StvP2 catalyzes the MDB formation in streptovaricins, with an atypical substrate inhibition kinetics. Furthermore, X-ray crystal structures in complex with substrate and structure-based mutagenesis reveal the intrinsic details of the enzymatic reaction. The mechanism of MDB formation is proposed to be an intramolecular nucleophilic substitution resulting from the hydroxylation by the heme core and the keto-enol tautomerization via a crucial catalytic triad (Asp89-His92-Arg72) in StvP2. In addition, in vitro reconstitution uncovers that C6-O-methylation and C4-O-acetylation of streptovaricins are necessary prerequisites for the MDB formation. This work provides insight for the MDB formation and adds evidence in support of the functional versatility of P450 enzymes.

---

[1] Key Laboratory of Combinatorial Biosynthesis and Drug Discovery, Ministry of Education, Wuhan University School of Pharmaceutical Sciences, 430071 Wuhan, P. R. China. ✉email: zhengyu.zhang@whu.edu.cn; yhsun@whu.edu.cn

Drug-resistance has been an urgent and inevitable concern with vast abuse of antibiotics[1]. A variety of drug-resistant pathogens are emerging at a faster rate than ever before. *Staphylococcus aureus* is a common pathogen in clinic, and methicillin-resistant *S. aureus* (MRSA) and even vancomycin-resistant *S. aureus* (VRSA) have already threatened human health seriously[2]. As a consequence to deal with the increasing number of drug-resistant pathogens, improved drug development strategies are urgently required.

Natural products are important sources of drug candidates and play a highly significant role in the drug discovery and development[3] due to their diverse chemical structures and biological activities. In addition to the basic scaffolds constructed with various simple building blocks, the complexity of natural products arise mainly from post-decorating enzymes to functionalize the biosynthetic skeleton[4]. Of which cytochrome P450 enzymes are one of the most exquisite and versatile biocatalysts found in nature because of the immense variety of substrate structures and the types of reactions they catalyze. They are a superfamily of heme-containing monooxygenases, and are involved in more than twenty different types of chemical oxidation reactions[4,5] with high chemo-, regio-, and stereoselectivity under mild conditions.

In our previous discovery for anti-MRSA bioactive natural products, streptovaricin C (**1**) and its derivatives (Fig. 1a, Supplementary Figs. 1–4 and Supplementary Table 1) were identified from *Streptomyces spectabilis* CCTCC M2017417 isolated from the soil in the campus of Wuhan University, and exhibited significant bioactivities[6]. The biosynthetic investigation of streptovaricin C in

this study might provide the foundation for future development of potent anti-MRSA drugs through biosynthesis-based structure modifications or chemical semi-synthesis. As a member of naphthalenic ansamycins, streptovaricins were gradually identified since 1957[7–9]. Remarkably, streptovaricin C is structurally related to rifamycins[10] and chemically semi-synthesized rifamycin derivatives (such as rifampicin, rifabutin, and rifapentine) that have been widely used for treatment of tuberculosis and infections caused by drug-resistant pathogens in clinics. However unlike the well-studied rifamycins[11], the biosynthesis of streptovaricin C especially the mode of formation and the timing of the unique naphthalene ring-coupled methylenedioxy bridge (MDB) moiety related to bioactivity remain unclear yet.

Here, we demonstrate that reduced form of damavaricin C as a vital precursor is finally transformed into MDB-contained mature streptovaricins (streptovaricin C and streptovaricin B or J, Supplementary Fig. 1) by cytochrome P450 enzyme StvP2 after sequential methylation and acetylation. Meanwhile, we reveal the underlying mechanism for MDB formation using crystal structure of StvP2 in complex with its substrate.

## Results

**StvP2 is responsible for the MDB formation of streptovaricins.** MDB is a characteristic moiety mainly existing in phenylpropanoids such as (−)-pluviatolide[12], (+)-pluviatilol[13,14], (+)-sesamin[13], etc. and alkaloids such as (S)-cheilanthifoline[15], (S)-canadine[16,17], (S)-stylopine[18], etc. from higher plants and also in some microbial-

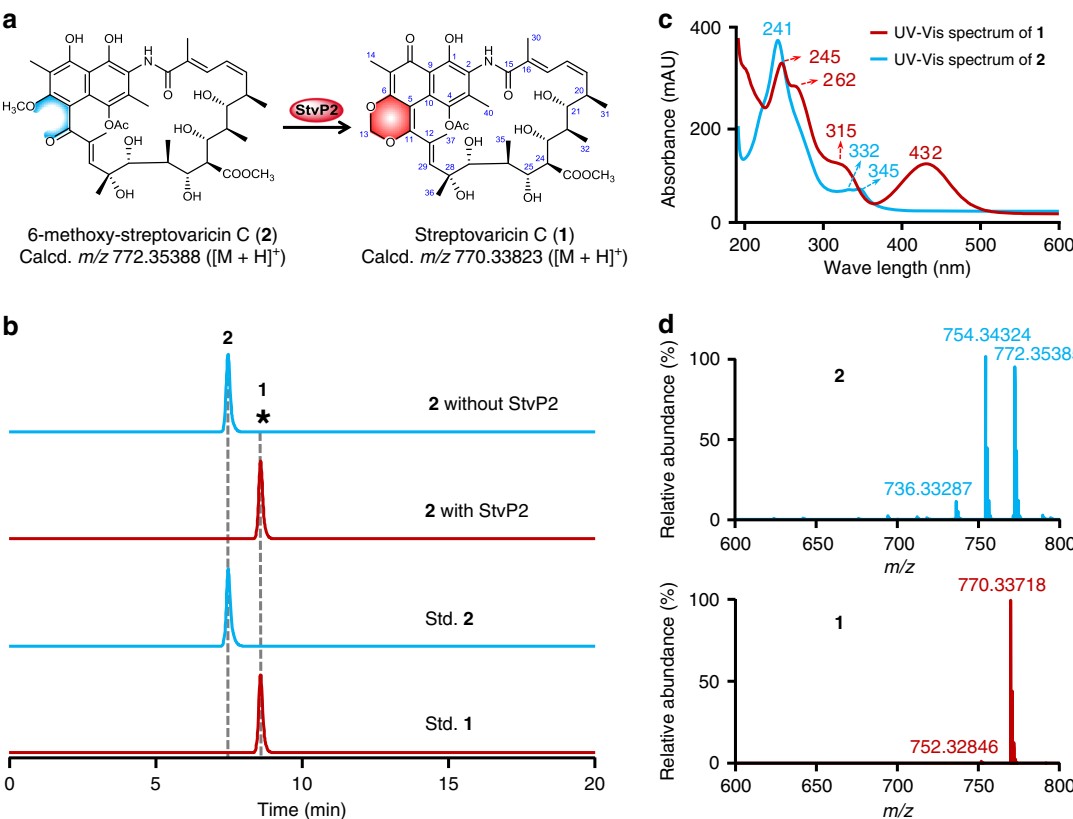

**Fig. 1 In vitro enzymatic conversion of 2 to 1 catalyzed by StvP2. a** Biochemical reaction formula of StvP2-catalyzed conversion of **2** to **1**. **b** LC–ESI–HRMS analysis of enzymatic conversion of **2** to **1** by StvP2. The asterisk indicated that no component corresponding to **1** was detected. **c** UV/Vis spectra of **1** and **2**. **d** ESI–HRMS spectra of **1** (calcd. $m/z$ 770.33823 [M + H]$^+$; 752.32767 [M + H − H$_2$O]$^+$) and **2** (calcd. $m/z$ 772.35388 [M + H]$^+$; 754.34332 [M + H − H$_2$O]$^+$; 736.33275 [M + H − 2H$_2$O]$^+$) extracted from total ion chromatography. The pure compounds **1** and **2** identified by NMR were used as standards (Std. **1** and Std. **2**). Experiments in **b**–**d** are representative of three independent experiments.

produced secondary metabolites[6,19]. According to our previous investigation of all five P450s in streptovaricins biosynthetic gene cluster[6], *stvP2* is the most promising candidate for MDB formation since the phylogenetic analysis of StvP2 with other identified MDB-formation related cytochrome P450s showed the closest relationship with XanO2 (Supplementary Fig. 5), a cytochrome P450 enzyme for *Streptomyces*-produced xantholipin[19]. However, the enzymatic conversion of protostreptovaricin I purified from ΔstvP2 by the recombinant StvP2 failed to form MDB-contained streptovaricins in our previous investigation[6]. To validate the hypothesis above, we expressed and purified the recombination 6 × His-tagged StvP2 (Supplementary Fig. 6) in *Escherichia coli* BL21(DE3), and conducted the StvP2-catalyzed in vitro assay using the whole crude extract from *stvP2* in-frame deletion mutant[6] as substrate. Strikingly, a new peak corresponding to **1** was detected by HPLC-DAD (high performance liquid chromatography-diode array detector) and confirmed by LC–ESI–HRMS (liquid chromatography–electrospray ionization–high resolution mass spectrometry) (Supplementary Fig. 7), but unexpectedly, no peak disappeared and diminished visibly. Based on the structure of **1** and the feature of MDB formation reactions, we speculated that 6-methoxy-streptovaricin C (**2**) was likely to be a reactive substrate for StvP2. To demonstrate this speculation, about 45 mg of 6-methoxy-streptovaricin C (**2**) as a most promising candidate was purified from ΔstvP2 fermentation and characterized by LC–ESI–HRMS (Supplementary Fig. 8) and NMR (nuclear magnetic resonance) (Supplementary Figs. 9–14 and Supplementary Table 1). The UV/Vis absorption spectrum of **2** obviously differs from **1** (Fig. 1c), which is probably due to the electron density change of the chromophore after MDB formation. Pure **2** was then incubated with StvP2 at 28 °C for 3 h in the presence of reductase pair ferredoxin (Fdx), ferredoxin-NADP⁺ reductase (FdR) as bio-reducing reagents along with NADPH. As a result, a component whose retention time, UV/Vis (ultraviolet–visible) spectrum and *m/z* value are well consistent with **1** was produced (yield > 98%) giving a direct evidence for the function of StvP2 to catalyze MDB formation in mature streptovaricins (Fig. 1). In addition, based on the fact in above biochemical results that oxidations at C-24 and C-28 have existed in both substrate and product, we considered that hydroxylation of C-28[6] and oxygenation of C-24 methyl group[6], should occur prior to the StvP2-catalyzed MDB biosynthetic process.

**StvP2 exhibits atypical substrate-inhibition kinetics.** To investigate the catalytic feature of StvP2, more putative substrates without MDB moiety were achieved through multiple gene knock outs. Thus, the triple mutants ΔstvP1P5P2 and ΔstvP1P4P2 were constructed on the basis of double mutants ΔstvP1P5 and ΔstvP1P4 (Supplementary Figs. 15, 16), in which both accumulated MDB-contained streptovaricins, i.e., streptovaricin H (**3**) with a hydroxyl group at C-28 (Fig. 2a(iii), b) and streptovaricin D (**5**) with a methoxycarbonyl side chain at C-24 (Fig. 2a(v), b), respectively. As predicted, both triple mutants ΔstvP1P5P2 and ΔstvP1P4P2 abolished the production of the mature MDB-contained streptovaricins, while the expected intermediates without MDB moiety, 6-methoxy-streptovaricin H (**4**) (Fig. 2b(iv), b) and 6-methoxy-streptovaricin D (**6**) (Fig. 2b(vi), b), were detected in their crude extracts by LC–ESI–HRMS. These compounds were purified from the fermented extracts on a large scale and their structures were determined by ESI–HRMS and NMR (Supplementary Figs. 14, 17–40 and Supplementary Tables 2, 3). It is worth noting that these substrates **4** and **6** can be converted by StvP2 in vitro to form the corresponding streptovaricins **3** and **5** with MDB moiety, exhibiting promiscuous substrate specificity of StvP2 (Supplementary Figs. 41, 42).

To intensively characterize the catalytic feature of StvP2 for substrates **2**, **4**, and **6**, the kinetic curves and parameters were measured. All kinetic curves of StvP2-catalyzed conversion of three substrates did not exhibit the classical Michaelis–Menten kinetics but the atypical substrate-inhibition kinetics (Fig. 3b) as some cytochrome P450 enzymes performed[20]. Considering that our attempts for fitting the kinetic curves with common substrate-inhibition kinetic models (the empirical substrate-inhibition kinetic model[21] and mechanistic two-site substrate-inhibition model[22]) were failed, we then tried the modified Hill equation[23,24] and all of three kinetic curves were fitted well with a good correlation coefficient ($R^2 > 0.95$) (Fig. 3b, Supplementary Table 4). Based on the kinetic parameters from the curves fitted, the $V_{max}$ (19.82 min⁻¹, corresponding to the turnover number $k_{cat}$) of **2** converted by StvP2 was the fastest amongst the three substrates, followed by that of **4** ($V_{max}$ for **4**, 14.45 min⁻¹, $k_{cat}$), and **6** ($V_{max}$ for **6**, 10.98 min⁻¹, $k_{cat}$), respectively. In addition, the $K_s$ and $K_i$ were comparable to $[S]_{1/2}$ value for the ascending and descending arm of the curves, respectively. The $V_{max}/K_s$ (0.18, corresponding to $k_{cat}/K_m$) of **2** was higher than that of **4** ($V_{max}/K_s$, 0.15) and **6** ($V_{max}/K_s$, 0.16), indicating the catalytic efficiency of **2** by StvP2 was higher than that of **4** and **6**. Taken together, intermediate **2** is the most favorable substrate for StvP2, which is also in accordance with the fact that streptovaricin C, as the direct product of **2** catalyzed by StvP2, is the major product in wild-type strain.

**StvP2 structure and its catalytic mechanism for MDB formation.** To enhance our understanding of MDB formation by StvP2, we solved the X-ray crystal structure of StvP2 at 1.35 Å resolution and the StvP2 in complex with substrate **2** at 2.3 Å resolution, respectively (Supplementary Table 5). The overall crystal structures of substrate-free and substrate-bound StvP2 adopts a typical P450 fold[4], consisted of twelve α-helices (A–L) and two β-sheets (1–2), with a heme as the core cofactor (Fig. 4a). A conserved cysteine residue (Cys349), positioned in the loop preceding L-helix, is coordinated to the iron of the heme cofactor as the proximal heme ligand, and a water molecule is also bound to the heme iron at a distance of 2.6 Å.

In substrate-bound StvP2 structure, a highly conserved "acid-alcohol" pair (Asp242-Ser243) (Fig. 4b) is positioned at a kink formed in the middle of the I helix (Asp227-Leu256), which is commonly a Glu/Asp-Thr pair (Ser could be considered as the one less carbon analog of Thr) in most P450 enzymes and controls the protonation of intermediate oxygen species during oxygen activation of P450 enzyme catalytic cycle[25,26]. Although there are some different modifications (such as Glu-Ser, Glu-Ala, Asp-Ala, Asp-Asn, etc.) in other P450 enzymes[26], these would be due to the unusual reactions catalyzed by the particular P450 enzymes. No electron density is present for the several residues of BC loop (Gly80-Arg81) and FG loop (Arg174-Ala187) (Fig. 4a), indicating that these regions are significantly flexible. The I helix, BC loop, the loop following the K helix, the second β strand of β 1 sheet and C-terminal residues together constructs a large substrate binding cavity by comparison of substrate-free and substrate-bound structures. The conformation of substrate-bound StvP2 structure showed no significant changes relative to the substrate-free StvP2 structure. After molecular replacement and refinement, the $F_O$-$F_C$ and $2F_O$-$F_C$ map, which are respectively generated from refinement results without and with substrate **2** model, are displayed in Supplementary Fig. 44. The high resolution and clear electron density of substrate **2** revealed its high occupancy in the crystal, which also facilitated greatly in substrate **2** molecule model building.

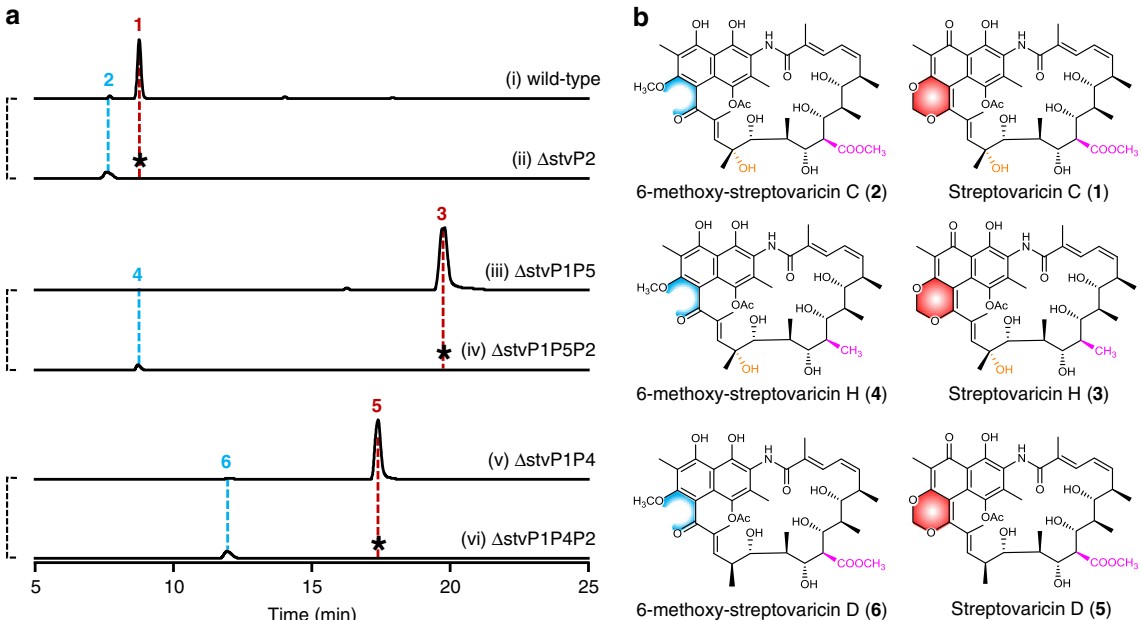

**Fig. 2 Comparative analysis of streptovaricins from wild-type strain and related mutants. a** LC–ESI–HRMS analysis of streptovaricins produced by wild-type strain and related mutants. The traces were extracted at calcd. $m/z$ ([M + H]$^+$) 770.33823, 772.35388, 726.34840, 728.36405, 754.34332, and 756.35897 corresponding to compound **1**, **2**, **3**, **4**, **5**, and **6**, respectively. The asterisks indicated no corresponding compounds were detected. **b** Chemical structures of streptovaricins from wild-type strain and related mutants. The structural differences among them are indicated with different colors. The MDB moieties are filled with red hexagons; their corresponding uncyclized parts are indicated with blue; different substitutions at C24 and C28 are indicated with magenta and orange, respectively. Experiments in **a** are representative of three independent experiments.

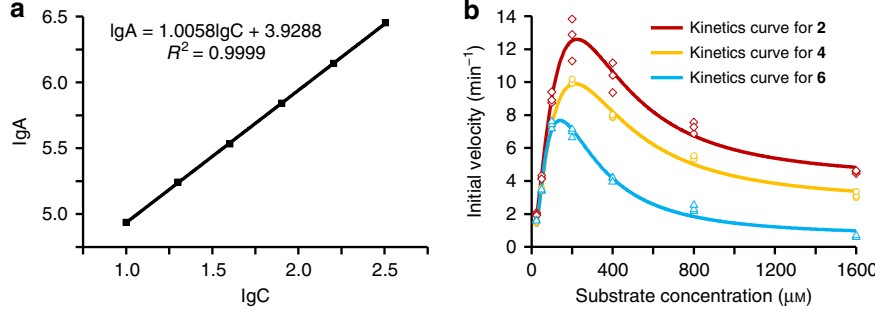

**Fig. 3 Substrate-inhibition kinetics of StvP2 catalyzing MDB formation. a** Calibration curve of **3** by HPLC-DAD with the peak area (A) as a function of substrate concentration (C) for quantification. **b** Kinetic curves of StvP2 converting **2**, **4**, and **6** to form **1**, **3**, and **5**, respectively, fitted with modified Hill model by Origin. Initial velocities were expressed with product formation relative to 1 μM enzyme. All the enzymatic kinetic experiments are conducted independently in triplicate ($n = 3$), and all the independent data points are plotted with red diamonds, yellow circles, and blue triangles respectively in **b**. Source data are provided as a Source data file.

The substrate molecule is almost perpendicular to the heme plane in the active pocket. There are multiple interactions between the substrate and the residues of the StvP2. The NH$_2$ group of the amide in Asn69 is hydrogen bonded to the carbonyl O atom of the amide bond in the substrate. Both of the NH of indole ring in Trp74 and the carbonyl O atom in Leu388 are hydrogen bonded to a water molecule, and this water molecule is further bond to the OH group at C21 of the substrate. In addition, the NH$_2$ group and the O atom of carbonyl in Ile390 are bond to the H atom of hydroxyl at C-23 and C-27, respectively (Fig. 4b). These interactions are contributed to bind the substrate molecule to a reasonable conformation to facilitate the enzymatic reaction. The methoxy group at C-6 in the substrate is pointed toward the iron center of the heme, which is essential to the monooxygenation by P450 enzyme. Moreover, a distinctive catalytic triad (Asp89-His92-Arg72)[27,28] observed in the BC loop (Fig. 4b) is considered as the crucial catalytic sites responsible for the

important keto-enol tautomerization for further MDB formation, which is never reported so far in this type of reactions and P450 enzymes to the best of our knowledge. Specifically, the carboxyl group of Asp89 is hydrogen bonded to the N-2 atom of imidazole group in His92, and meanwhile, the N-4 atom of imidazole group in His92 is hydrogen bonded to the O atom of the water molecule, which further forms a bifurcated hydrogen bond with the hydroxyl group at the C-1 of the substrate and the imine group of the guanidyl in Arg72, respectively. These three amino acid residues are almost in a straight line, approximately facing and paralleling with the naphthol plane. According to these crucial interactions, a reasonable catalytic mechanism is proposed to reveal the formation process of MDB moiety in the streptovaricins (Fig. 5a).

The carboxyl group of Asp89 is hydrogen bonded to the imidazole of His92 to protonate the N-2 atom of the imidazole group, which further provided a proton to the water molecule,

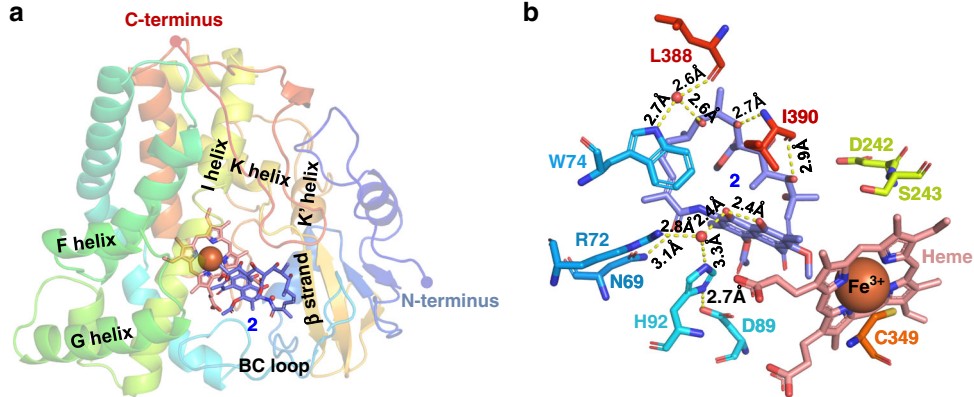

**Fig. 4 Structure of StvP2 with substrate 2. a** Structure of StvP2 with **2**. The structure is presented in rainbow. The conserved secondary structure of P450 in StvP2 is labeled in the structure, including BC loop and FGIK helices. N-terminus (blue) and C-terminus (red) are labeled as ball. Heme is colored in salmon and its $Fe^{3+}$ ion is colored in orange. The substrate **2** is colored in blue. **b** Interactions of **2** with the residues in StvP2. The residues within 5 Å around the substrate that constitute a large substrate binding pocket are labeled out. The dashed yellow lines represent the H-bond interactions between substrate **2** and the amino acid residues in the active pocket. Water molecules involved in the catalytic process of the triad are shown with red spheres.

**Fig. 5 Proposed catalytic mechanism of StvP2 catalyzing MDB formation. a** Proposed mechanism of MDB formation catalyzed by StvP2. **b** HPLC-DAD analysis at $\lambda_{max}$ 245 nm of enzymatic conversion of **2** to **1** by StvP2 site-directed mutants. The asterisks indicated no corresponding compounds were detected. The pure compounds **1** and **2** identified by NMR were used as standards (Std. **1** and Std. **2**). The results in **b** are representative of three independent experiments.

and in turn, the protonated water molecule forms a hydrogen bond with the imine group of the guanidyl in Arg72 to protonate the guanidyl moiety. Thus, a proton is transferred from Asp89 to Arg72 mediated by the His92 and a water molecule. Then hydroxyl group at C-1 of the substrate is hydrogen bonded to the water involved above, and meanwhile, this water molecule also forms a hydrogen bond with the N-4 atom of imidazole in His92 to protonate the imidazole group, which provides a proton to the carboxyl in the Asp89 to restore their undissociated state for next round of catalytic reactions. The proton of C-8 hydroxyl group is transferred to the O atom linked to C-1 of the substrate due to the dissociation of the hydroxyl caused by the catalytic process above, which triggers the keto-enol tautomerization to form an oxygen anion at C11. Afterwards, this reactive oxygen anion attacks the electrophilic center of protonated hemiacetal moiety, transformed by the heme active center of P450 enzyme and further protonated, through the nucleophilic substitution mechanism to form the MDB group, i.e., the substrate **2** is transformed to the product **1** containing MDB group.

To gain further insight into the catalytic mechanism the triad involved, site-directed mutations were introduced at these crucial amino acid residues (H92A, H92D, H92K, D89A, D89E, D89K, R72A, R72E, and R72K). All His92-mutated StvP2 mutants (H92A, H92D, and H92K) completely abolished the catalytic ability as expected (Fig. 5b(ii)–(iv)) due to its special ability of proton acceptor and donor. Based on the catalytic mechanism above, the amino acid at position 89 should be an acidic residue to transfer the proton to the Arg72 via the water molecule and the His92, and meanwhile, to isomerize the His92. In the biochemical assay of these three Asp89-mutated StvP2 mutants, only D89E retained a little weaker catalytic activity (Fig. 5b(vi)) than the wild-type StvP2 (Fig. 5b(xi)) for MDB formation. Both D89A and D89K mutants cannot provide the crucial proton to complete the proton transfer and lose the catalytic ability (Fig. 5b(v), (vii)). This proved the importance of this acidic residue. The residue at position 72 should be a basic amino acid to accept the proton from the proposed catalytic mechanism. In vitro conversions of these three Arg72-mutated StvP2 mutants showed the similar results (Fig. 5b(viii)–(x)) as that of Asp89-mutated StvP2 mutants, demonstrating the necessity of the basic residue at position 72. Furthermore, based on the amino acid sequence alignment of StvP2 with other cytochrome P450 enzymes (Supplementary Fig. 45), the amino acids corresponding to Asp89, His92, and Arg72 of StvP2 are conserved in their own subfamily (CYP81Q and CYP719A), but obviously different from the residues in StvP2. This indicated that residues of these positions would also be very crucial to the function of StvP2. These biochemical evidences and sequence alignment further supported the catalytic mechanism proposed above and the role of these catalytic residues.

The MDB moiety of streptovaricins in this work was found to form between a methoxy and a keto group, unlike the reported MDB moieties[12–19] in natural products to date. This was almost impossible from the principles of chemistry, unless some changes, such as the reduction of the keto or the tautomerism of the keto to a hydroxyl, occur. This very special structure characteristic of substrate **2** indicates the unique reaction mechanism, which is in accordance with our finding that these catalytic residues did not conserve in other MDB forming cytochrome P450 enzymes (Supplementary Fig. 45).

**The 6-*O*-methylation is essential for MDB formation.** Based on above results of StvP2-catalyzed MDB formations in streptovaricins and proposed MDB formation mechanism, the methyl group of C6–OH on the naphthalene ring appeared to provide a

linker carbon atom of two oxygen atoms in the MDB moiety, and it ought to be catalyzed by an *O*-methyltransferase. According to previously conducted bioinformatics analysis, three methyltransferase encoding genes exist in the streptovaricin biosynthetic gene cluster[6]. Of these three methyltransferases, StvM1 shares high similarity with the class I *S*-adenosyl-methionine(SAM)-dependent methyltransferase in *Streptomyces* sp. NRRL B-1347, indicating that StvM1 might be responsible for methylation of C6–OH in streptovaricins. To verify the function of StvM1, the *stvM1* in-frame deletion mutant was constructed (Supplementary Fig. 46), in which all the mature MDB-containing streptovaricins, such as streptovaricin C and G, etc. were abolished, instead, an intermediate was accumulated (Fig. 6a(iv)) and its chemical structure was determined as damavaricin C (**7**) by ESI–HRMS (Supplementary Fig. 47) and NMR (Fig. 6b, c and Supplementary Figs. 14, 48–52 and Supplementary Table 6). Such phenotype of ΔstvM1 that loss of products with MDB moiety and missing methyl group of C6–OH in **7** showed a reasonable correlation and timing between methylation of C6–OH and MDB formation. This conclusion was further confirmed by in vitro assay, in which the recombinant StvM1 (Supplementary Fig. 6) successfully converted **7** to 6-methoxy-damavaricin C (**8**, isolated from ΔstvA2 as standard here, Supplementary Figs. 14, 53–58 and Supplementary Table 6) using SAM as methyl donor (Fig. 7a(iii)).

**The 4-*O*-acetylation is necessary for MDB formation.** According to the substrate structure in the enzymatic conversion of StvP2-catalyzed MDB formations, the *O*-acetylation at C-4 of streptovaricins is formed prior to the MDB, giving us a valuable hint that it may be a premise for the MDB biosynthesis. Interestingly, there are two acyltransferase encoding genes, *stvA1* and *stvA2*, in the streptovaricin biosynthetic gene cluster, however three *O*-acetylations at C4, C21, and C-25 are required based on all identified streptovaricin structures[6]. Apparently, *stvA1*, which is 10.1 kb away from the cluster, could be ruled out due to the lack of sequence similarity and the unusual short nucleotide length (390 bp) compared with known acyltransferase[29,30] encoding gene (~1.2 kb). To interrogate the function of these two genes, each of them was knocked out by in-frame deletion, respectively (Supplementary Figs. 59, 60). According to the HPLC-DAD and LC–ESI–HRMS analysis, the production of all MDB-contained streptovaricins were disrupted in ΔstvA2 but two major components **7** and **8** without any *O*-acetylation at C-4, C-21, and C-25, and MDB moiety either, were accumulated (Fig. 6a(iii), b–d). In contrast, for ΔstvA1, there is no difference with wild-type in streptovaricins production (Fig. 6a(ii)) and no apparent involvement in streptovaricins biosynthesis. These indicated that StvA2 should be responsible for more than one *O*-acetylation with different substrates during streptovaricins biosynthesis.

To explore the function of StvA2 in MDB biosynthesis, in vitro assays were conducted in the presence of acetyl CoA as donor of acetyl group. To our surprise, neither of intermediates **7** and **8** were catalyzed by recombinant StvA2 (Fig. 7 and Supplementary Fig. 61) to give *O*-acetylation at C-4 of the naphthalene ring. As for the enzymatically catalytic feature of acetyltransferase, none has been reported up to date to function via directly adding an acetyl group to an unsaturated carbonyl group in the naphtho-quinone moiety. An acetyltransferase usually catalyzes an acetyl group provided by the acetyl CoA to link to a nucleophilic group, such as hydroxyl group[31,32], amino group[33,34], sulfhydryl group[35], etc. through nucleophilic attack mechanism. Recently, a super-ficially similar conversion from rifamycin S to rifamycin L during rifamycin biosynthesis showed a transfer of a hydroxyacetyl group to C-4 keto catalyzed by Rif15a/b, a transketolase without any similarity with acetyltransferase, using a fructose-6-phosphate

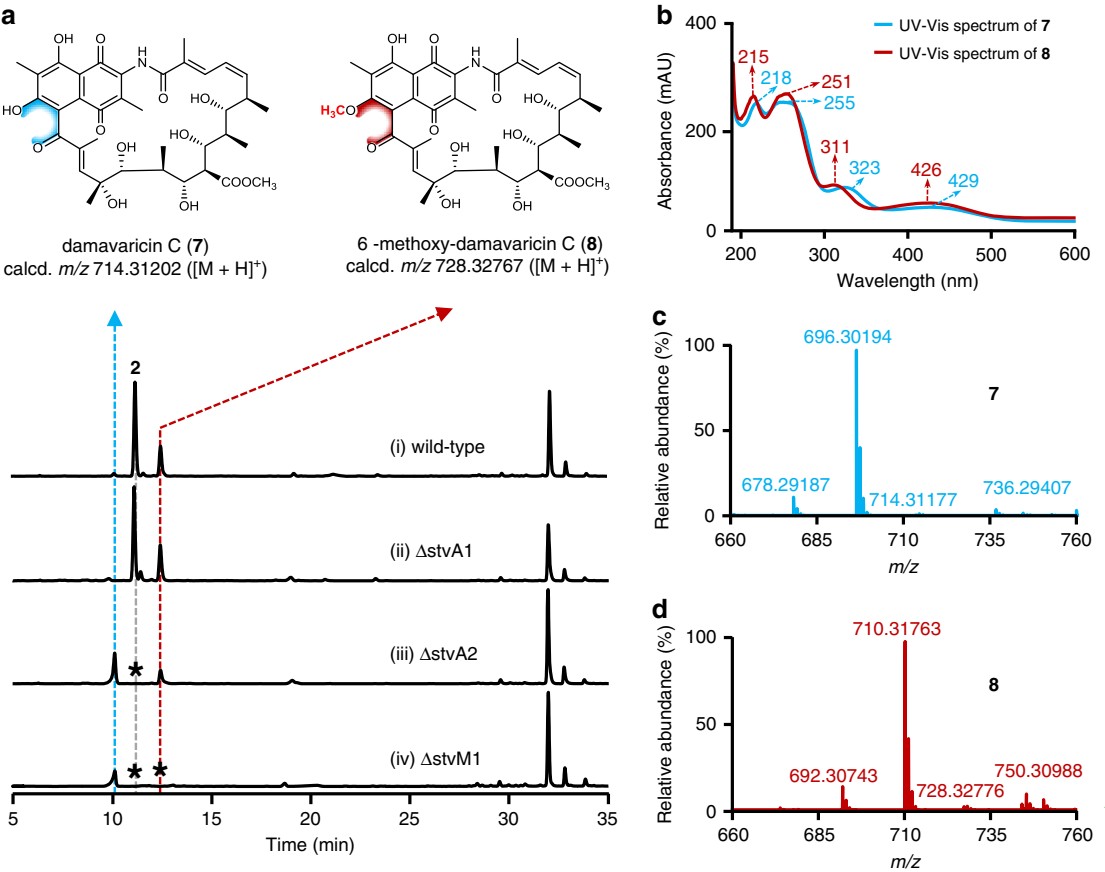

**Fig. 6 Comparative analysis of streptovaricins from wild-type strain and related mutants. a** HPLC-DAD analysis at $\lambda_{max}$ 432 nm of streptovaricins from the wild-type strain and related mutants ΔstvA1, ΔstvA2, and ΔstvM1. It is noted that the peak of **8** was overlapped with another one in wild-type and ΔstvA1 due to the almost same retention time. The asterisks indicated no corresponding compounds were detected. **b** UV/Vis spectra of **7** and **8**. **c**, **d** displayed ESI–HRMS spectra of **7** and **8**, respectively. ESI–HRMS spectra of **7** (calcd. $m/z$ 714.31202 [M + H]$^+$; 736.29396 [M + Na]$^+$; 696.30145 [M + H − H$_2$O]$^+$; 678.29089 [M + H − 2H$_2$O]$^+$) and **8** (calcd. $m/z$ 728.32767 [M + H]$^+$; 750.30961 [M + Na]$^+$; 710.31710 [M + H − H$_2$O]$^+$; 692.30654 [M + H − 2H$_2$O]$^+$) extracted from total ion chromatography. The results in **a–d** are representative of three independent experiments.

as a keto donor[36]. It hints that the authentic substrate for the acetyltransferase StvA2 to form *O*-acetylation on naphthalene ring could be a naphthol structure instead of a naphthoquinone. Moreover, the naphthol structure was reported to be readily oxidized to a naphthoquinone form by the aerial oxygen or divalent metal ions, such as Mn$^{2+}$, Cu$^{2+}$, Co$^{2+}$ etc[37]. So we assume when the naphthol structure is produced during the biosynthesis of streptovaricins, if not immediately protected by acetylation on the naphthol hydroxyl, the naphthol structure would be subsequently oxidized to a naphthoquinone inside the cell or after released to the environment outside the cell. Based on it, we speculate that **7** and **8** isolated from ΔstvA2 are very likely an oxidized form of the initial substrates, which might be the reason for the failure of above in vitro assay. They can be used as authentic substrate for StvA2 to produce the acetylation products only when they are in a reduced form. To test it, Fdx and FdR from the energy transfer system of chloroplast in *Spinacia oleracea*[38] were applied to in vitro assay. Incubation of these reductase pair and NADPH with StvM1, StvA2, substrate **7** at 28 °C for 3 h successfully led to the formation of expected product **2** (Fig. 7a(vii)), and even more amounts of the compound with double acetylations at C-4 and C-21 or C-25, compared with negative control without Fdx and FdR (Fig. 7a(vi)). It demonstrated the indispensability of reduced intermediate during MDB formation. Meanwhile, this set of results also provided an evidence for the order of StvM1-catalyzed methylation of C6–OH and StvA2-catalyzed acetylations because the acetylation reaction

cannot take place without StvM1 (Fig. 7a(iv, v)), while methylation must happen no matter whether StvA2 exists or not (Fig. 7a (ii, iii, vi, viii, ix)). Obviously, the methylation of C6–OH became a prerequisite for *O*-acetylation of C-4 by StvA2. But afterwards there exist a cross network instead of sequential steps between MDB formation and the rest acetylations at C-21 or C-25 (Fig. 7b and Supplementary Fig. 62). To further confirm it, we used **2** as direct substrate to carry on the enzymatic conversion by StvA2 and/or StvP2. The results convinced that two parallel pathways for MDB formation and acetylations at C-21 or C-25 (Supplementary Fig. 63). It is noteworthy that, against our expectation, no detectable product with *O*-acetylation of C-4 and the rest acetylations was found when **8** was used as substrate in the above reaction system of **7** (Supplementary Fig. 61). The possible reason could be that the methylation of C6–OH, which is the only difference between **7** and **8**, hindered the recognition of ferredoxin with this molecule.

Curiously, in contrary to the wild-type, we found that the yield of 21 or 25-acetoxy-streptovaricin C was significantly higher than that of **1** in the result of in vitro reconstitution (green and red peak in Fig. 7a(ix), respectively). The possible reason could be the catalytic efficiency of StvA2 for substrate **2** and **1** under the unnatural condition is higher than that in vivo, without strict compartmentalization and regulation by certain mechanism in vitro. In addition, another irrational phenomenon was observed in above in vitro reconstitution. As equivalent intermediates in a parallel pathway driven by StvA2 and StvP2, **1** had obvious

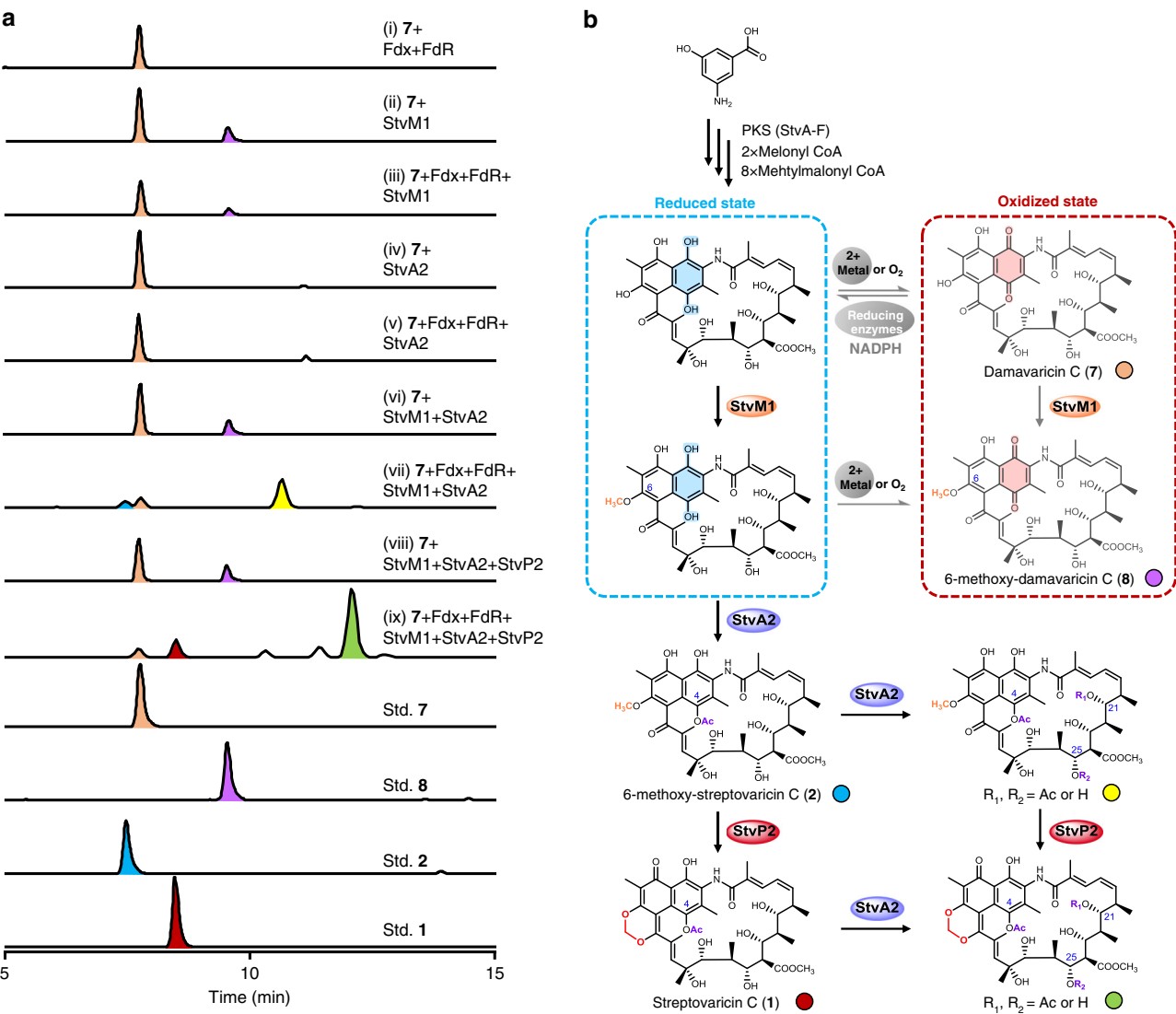

**Fig. 7 In vitro enzymatic conversions of compound 7 and proposed biosynthetic pathway of MDB formation in streptovaricins. a** LC–ESI–HRMS analysis of enzymatic conversion by StvM1 and/or StvA2 and/or StvP2 using **7** as substrate. (i) **7** in reaction buffer with Fdx, FdR, and NADPH. (ii) **7** and StvM1 in reaction buffer without Fdx, FdR, and NADPH. (iii) **7** and StvM1 in reaction buffer with Fdx, FdR, and NADPH. (iv) **7** and StvA2 in reaction buffer without Fdx, FdR, and NADPH. (v) **7** and StvA2 in reaction buffer with Fdx, FdR and NADPH. (vi) **7**, StvM1 and StvA2 in reaction buffer without Fdx, FdR, and NADPH. (vii) **7**, StvM1 and StvA2 in reaction buffer with Fdx, FdR, and NADPH. (viii) **7**, StvM1, StvA2, and StvP2 in reaction buffer without Fdx, FdR, and NADPH. (ix) **7**, StvM1, StvA2, and StvP2 in reaction buffer with Fdx, FdR, and NADPH. All the enzymatic reactions were incubated at 28 °C for 3 h. Pure compounds (**1**, **2**, **7**, and **8**) identified by NMR were used as standards or substrate in assay. The traces were extracted at calcd. *m/z* for [M + H]$^+$: 770.33823 (**1**, red peak), 772.35388 (**2**, blue peak), 728.32767 (**8**, purple peak), 714.31202 (**7**, pink peak), 814.36445 (yellow peak indicated the compound with double acetylations at C-4 and C-21, or C-25), and 812.34880 (green peak indicated the MDB-contained compound with double acetylations at C-4 and C-21 or C-25). **b** A proposed biosynthetic pathway of MDB formation in streptovaricins. Experiments in **a** are representative of three independent experiments.

accumulation but no 21 or 25-acetoxyl-6-methoxy-streptovaricin C was detected (Fig. 7a(ix)). This might be due to the difference of catalytic efficiency between StvA2 and StvP2, which was supported by time dependent conversion reactions (Supplementary Fig. 64). The intermediate **1** transformed from **2** by StvP2 was accumulated due to the lack of timely conversion of StvA2. While, the comparable intermediate 21 or 25-acetoxyl-6-methoxy-streptovaricin C from **2** by StvA2 could not be accumulated to be detected owing to the efficient transformation of StvP2.

In view of the above results, we reconstituted MDB biosynthetic pathway in vitro. Incubation of substrate (**7**) and proteins (StvM1, StvA2, and StvP2) tested together with reductive cassette (Fdx, FdR,

and NADPH) efficiently yielded 25 or 21-acetoxy-streptovaricin C (i.e., streptovaricin B or J, respectively, Supplementary Fig. 1) and less amount of **1** both containing MDB (Fig. 7a(ix)). These products did not form without either StvP2 or reductive cassette (Fig. 7a(vii, viii)).

In conclusion, we have identified StvP2 as a key enzyme which is responsible for MDB formation, and determined the timing of its biosynthesis. We propose that StvM1 first catalyzes methylation of C6–OH, followed by the first acetylation on C4–OH by StvA2. Subsequently, either further acetylations or MDB formation occurs through two parallel pathways and finally converge to 25 or 21-acetoxy-streptovaricin C (Fig. 7b).

## Discussion

MDB is a crucial moiety shared in many natural products, and plays a vital role in their bioactivity. Deciphering the detailed catalytic mechanism of StvP2 for MDB formation in streptovaricins biosynthesis especially based on crystal structures is helpful to deepen our understanding of the biosynthesis of other similar natural products containing MDB. Moreover, enzymatic engineering can be used to produce more MDB-contained natural products. To our knowledge, the StvP2 crystal structure in complex with substrate **2** elucidated in this study is the first mechanistic characterization of a cytochrome P450 enzyme catalyzing MDB formation. Attempts to obtain product **1** bound StvP2 did not result in a complex crystal even using same or higher compound dose during crystal soaking. This indicates the different binding ability between StvP2 and **1** or **2**. Since **1** is the product of StvP2, the low binding ability of **1** and high binding ability of **2** could be recognized as the binding ability difference of substrate and product of an enzyme during a catalysis reaction. This feature can dramatically facilitate catalysis because substrate force product out of StvP2 easily.

During our tracing for the vital substrate of StvA2, the high instability of reduced intermediate (the reduced form of **7** and **8**) could not be obtained and only be isolated as the unnatural naphthoquinone intermediate (**7** and **8**) in oxidation state. These unavailable substrates became a big obstacle for further study. We cannot make a breakthrough until the reductive cassette was introduced. Initially, a series of bio-reducing reagents were attempted to this reduction process. It is reported that the crude protein extracts from *E. coli* K12 can utilize NADPH to reduce rifamycin S to rifamycin SV[37], indicating that certain endogenous proteins could have ability to reduce similar naphthoqinone to naphthol. Therefore, we tried the crude protein extract from alternative *E. coli* BL21(DE3), a strain derived from *E. coli* B with more than half of the encoded proteins identical to that of *E. coli* K12 strain[39], to reduce **7** and **8** for the subsequent reactions by StvM1 and StvA2. As expected, the results showed that **7** was successfully converted to **2** in vitro in the presence of StvM1 and StvA2 with a fairly low efficiency and some side products due to the crude protein extract used. To increase catalytic efficiency, four putative quinone-reductase encoding genes (*qorA*, *chrR*, *wrbA*, and *nuoK*) from BL21(DE3) were cloned and expressed to be used for reduction of naphthoquinone to naphthol. Unfortunately, all of these recombinant quinone-reductases failed to realize this reduction. However, a pair of reductases Fdx and FdR "borrowed" from *Spinacia oleracea* allowed us to successfully overcome this obstacle. Despite its own reduced enzymes from wild-type strain for this reduction remain undefined, but we believe that it should be highly homologous with what we "borrowed" from *Spinacia oleracea*, or in the other hand there is no need at all for such enzyme to convert the oxidized intermediate to reduced form inside the reductive environment of cell.

In brief, we have determined the function and catalytic characteristic of StvP2 through in vitro biochemical evidences and kinetic parameters measurement, and revealed the StvP2-catalyzed MDB formation mechanism during streptovaricin C biosynthesis in combination with X-ray crystal structures. Together with the function investigations of StvM1 and StvA2, we have reconstituted the core biosynthetic pathway of streptovaricins from damavaricin C to streptovaricin B or J in vitro. Deciphering of the enzymatic mechanism of StvP2-catalyzed MDB formation not only enriches the versatility of cytochrome P450 enzymes in catalyzing complicated natural products, but also contributes to the unknown MDB formation in other natural products. The discovery of the core streptovaricin biosynthetic pathway is a crucial step for elucidating the whole biosynthetic pathway of streptovaricins, and may also provide some insight to the biosynthesis of ansamycins in the same family. Moreover, the revelation of these catalytic mechanism and biosynthetic pathway provide the potential possibilities for the rational design and modification of anti-MRSA drugs based on the combinatorial biosynthesis and chemically semi-synthesis.

## Methods

**General methods.** Restriction endonucleases, Phusion High-Fidelity Master Mix with GC-buffer, and Gibson Assembly® Master Mix were obtained from New England Biolabs. T4 DNA ligase was purchased from Themo Fisher Scientific Co. Ltd. Oligonucleotide primers were synthesized by GenScript and Tsingke. DNA sequencing of PCR products was performed by GenScript or Tsingke. All chemical reagents and antibiotics were purchased from Sigma-Aldrich and Sangon Biotech.

*Streptomyces spectabilis* CCTCC M2017417 wild-type strain and its mutants were cultured at 28 °C in ABB13 solid medium (0.5% soluble starch, 0.5% tryptone soya broth, 0.21% MOPS, 0.0012% $FeSO_4 \cdot 7H_2O$, 0.001% thiamine hydrochloride, 0.3% $CaCO_3$, and 1.8% agar), and in TSBY liquid medium (3% tryptone soya broth, 0.5% yeast extract, 10.5% sucrose) at 220 rpm for 36 h to obtain the seed cultures for fermentation or intergeneric conjugation. *Streptomyces* strains were fermented in SFM liquid medium (3.3% soya flour, 2% mannitol) at 28 °C and 220 rpm for 5 days. *E. coli* strains were cultured at 37 °C in 2 × TY liquid or solid medium (1.6% tryptone, 1.0% yeast extract, 0.5% NaCl, and 2% agar for solid medium) at 37 °C with appropriate antibiotic selection at a final concentration of 25 µg mL$^{-1}$ apramycin, 25 µg mL$^{-1}$ chloramphenicol, and 50 µg mL$^{-1}$ kanamycin.

**HPLC-DAD analysis.** All samples extracted from fermentation and in vitro assay were preliminary analyzed by Shimadzu HPLC (LC-20AT) with a DAD detector (SPD-M20A) through Phenomenex analytic column (C18, 5 µm, 4.6 × 250 mm). The mobile phase A was 1‰ formic acid in water, and the mobile B was pure acetonitrile (HPLC grade). The flow rate was 1.0 mL min$^{-1}$. The gradient elution condition was as follows: Initially, the concentration of B was changed from 35 to 65% linearly in 20 min, next increased to 95% gradually from 20 to 28 min. The ratio of mobile phase B was then hold at 95% for 2 min. Finally, it decreased to 35% from 30 to 33 min and hold on for 5 min to make a balance.

**LC–ESI–HRMS analysis.** All samples extracted from fermentation and in vitro assay were further analyzed by Thermo Scientific HPLC equipped with ESI ion source and orbitrap high resolution mass analyzer using the same analytic chromatography column and gradient elution conditions as above. The MS scan range was set at *m/z* 300–1000.

**NMR analysis.** The NMR spectra were acquired on Bruker 400 or 600 NMR spectrometer at 298 K. All the NMR data were analyzed using MestRenova 9.0 software with the deuterated solvents as external standard, and reported as δ in ppm and *J* in Hz. Multiplicities were expressed as s (singlet), d (doublet), t (triplet), q (quartet), dd (double doublet), and m (multiplet), etc. All the signals were assigned by comprehensive analysis of [1]H NMR, [13]C NMR, DEPT135, HSQC, [1]H–[1]H COSY, and HMBC spectra.

**Preparation of recombinant proteins StvM1 and StvA2.** Two DNA fragments containing *stvM1* and *stvA2*, respectively, were amplified from the genome DNA of wild-type strain *S. spectabilis* CCTCC M2017417 under standard PCR conditions using the primers StvM1-For and StvM1-Rev (for *stvM1*), and StvA2-For and StvA2-Rev (for *stvA2*), respectively. The DNA fragments were purified with DNA gel extraction kit and then cloned into the site between *Nde*I and *Eco*RI of vector pET28a(+) by Gibson Assembly method to obtain the plasmids pET28a(+) +stvM1 and pET28a(+)+stvA2, respectively, which were verified by restriction endonuclease digestion and sequencing. The two plasmids were transformed into *E.coli* BL21(DE3) strain, respectively, for the N-terminal 6 × His-tagged recombinant protein expression of StvM1 and StvA2, respectively.

The *E.coli* BL21(DE3) transformant carrying the plasmid pET28a(+)+stvM1 and pET28a(+)+stvA2, respectively, were grown at 37 °C overnight in 2 × TY liquid medium supplemented with 50 µg mL$^{-1}$ kanamycin. The overnight seed culture was inoculated (1:100) into 1 L LB broth (1% tryptone, 0.5% yeast extract, and 1% NaCl) supplemented with 50 µg mL$^{-1}$ kanamycin, and then cultured at 37 °C until the $OD_{600}$ reached 0.8, when 0.2 mM isopropyl β-D-thiogalactoside (IPTG) was added to the liquid culture to induce protein expression at 18 °C for 18 h. The cells were collected by centrifugation at 5,000 × g for 10 min, and resuspended with 30 mL lysis buffer (20 mM Tris-HCl, 500 mM NaCl, 10 mM imidazole, and pH 8.0), followed by sonication (3 s on/5 s off) for 30 min on ice. The cell lysate was transferred into the centrifuge tubes (Beckman Coulter) and centrifuged at 40,000 × g for 1 h. The supernatant were loaded on a nickel-nitrilotriacetic acid resin column pre-equilibrated with lysis buffer, and eluted with 15 mL wash buffer (20 mM Tris-HCl, 300 mM NaCl, 20 mM imidazole, and pH 8.0), and each of 5 mL elution buffer with gradient concentration of imidazole (20 mM Tris-HCl, 300 mM NaCl, pH 8.0, with imidazole from 50 to 800 mM). The eluents were subjected to SDS-PAGE analysis and target protein-contained eluents

were combined, then ultra-filtrated and buffer-exchanged to storing buffer (25 mM Tris-HCl, 150 mM NaCl, 2 mM DTT (dithiothreitol), 10% glycerol, pH 7.5) using Amicon Ultra-15 centrifugal filter units (Millipore) with a 30 kDa cutoff. All the purification processes were performed at 4 °C.

**Preparation of recombinant protein StvP2.** For crystallization of StvP2, the construction of the recombinant protein expression plasmid was slightly modified, in which eight amino acids were removed between the end of N-terminal $6 \times$ His-tag and the beginning of StvP2. To achieve it, two DNA fragments were amplified from the pET28a(+) and the genome DNA of wild-type strain under standard PCR conditions using the primers StvP2-pet-his-1 and StvP2-pet-his-2 (for partial vector), and StvP2-pet-his-3 and StvP2-pet-his-4 (for StvP2), respectively, and connected with the linearized pET28a(+) by double digested with *Sph*I and *Eco*RI through Gibson Assembly to obtain the plasmid pET-His-stvP2. The plasmid was verified by restriction endonuclease digestion and sequencing. The plasmid pET-His-stvP2 was transformed into chemically competent *E.coli* C43(DE3) strain on LB plate containing kanamycin.

A single transformant harboring the pET-His-stvP2 plasmid was inoculated into 10 mL of LB medium containing 50 μg mL$^{-1}$ kanamycin, and cultured at 200 rpm and 37 °C overnight. The overnight seed culture was inoculated into 1 L of LB medium supplemented with 50 μg mL$^{-1}$ kanamycin and cultured at 200 rpm and 37 °C until the OD$_{600}$ reach 0.6–0.8, when the culture was cooled and supplemented with 0.1 mM IPTG, 400 μM ALA (5-aminolevulinic acid), and 200 μM (NH$_4$)$_2$Fe(SO$_4$)$_2$ to induce the expression of StvP2 for another 18 h.

After harvested by centrifugation at $5,000 \times g$ for 10 min, cells were suspended in TBS buffer (20 mM Tris, 300 mM NaCl, and pH 8.0) supplemented with 10% glycerol and 1 mM PMSF, followed by lysis through high pressure homogenizer and centrifugation at $40,000 \times g$ for 1 h to remove cell debris. The resulting supernatant was treated with 5 mL of HisTrap-HP column, and then further purified by gel-filtration equilibrated with TBS (20 mM Tris, 300 mM NaCl, and pH 8.0). The protein eluent was concentrated using Amicon 30,000 MWCO centrifugal filters, and the concentration was measured with NanoDrop.

**Preparation of site-directed mutant StvP2.** PCR-based mutagenesis was used to produce *stvP2* mutants encoding H92A, H92D, H92K, D89A, D89E, D89K, R72A, R72E, and R72K proteins. Each pair of two DNA fragments were amplified from the plasmid pET-His-stvP2 template with two pairs of primers, StvP2-pet-his-3 and reverse mutagenesis primer (H92A/H92D/H92K/D89A/D89E/D89K/R72A/R72E/R72K-Rev), and StvP2-pet-his-4 and forward mutagenesis primer (H92A/H92D/H92K/D89A/D89E/D89K/R72A/R72E/R72K-For) (Supplementary Table 9), respectively, followed by purification with DNA gel extraction kit, and then cloned into pET-His vector (double digested the pET-His-stvP2 plasmid with *Nde*I and *Eco*RI by Gibson Assembly kit. Plasmids were then isolated and sequenced to verify the desired mutation. The resultant plasmids were then transformed into the competent *E. coli* C43(DE3) cells for recombinant protein expression. The rest expression and purification process of StvP2 mutants were the same as the wild-type StvP2 protein described above.

**Preparation of recombinant protein Fdx and FdR.** A pair of reductase Fdx and FdR from *Spinacia oleracea* was used as the bio-reducing reagent for cytochrome P450 enzyme-catalyzed biochemical reactions. The plasmids[38] containing *fdx* and *fdR* genes were transformed into *E. coli* BL21(DE3) strain, respectively and the transformant was inoculated in $2 \times$ TY liquid medium supplemented with 50 μg mL$^{-1}$ kanamycin to grow at 37 °C overnight. The overnight seed culture was inoculated (1:100) into 1 L LB broth (1% tryptone, 0.5% yeast extract, and 1% NaCl) supplemented with 50 μg mL$^{-1}$ kanamycin, and then cultured at 37 °C until the OD$_{600}$ reached 0.8, when 0.1 mM isopropyl β-D-thiogalactoside (IPTG) was added to induce protein expression at 18 °C for 18 h. The rest expression and purification process of these proteins were the same as that of StvM1.

**Enzymatic conversion by wild-type or mutated StvP2.** In vitro biochemical reactions by wild-type or mutated StvP2 were carried out in a 100 μL reaction system containing 2.5 μM of StvP2 with 200 μM substrate (**2** or **4** or **6**), 12.5 μM Fdx, 2.5 μM FdR, and 2 mM NADPH in 50 mM Tris-HCl buffer (pH 7.5) at 28 °C for 3 h. Control assays were run alongside each reaction without StvP2. The reaction mixtures were then quenched with identical volume of methanol and centrifuged at the speed of $13,800 \times g$ for 5 min. The resulting supernatants were finally analyzed by HPLC-DAD and LC–ESI–HRMS.

**Enzymatic conversion of 1 or 2 by StvA2 or/and StvP2.** A 100 μL reaction system containing 100 μM of **1** or **2** with 2.5 μM StvA2, 200 μM acetyl CoA in 50 mM Tris-HCl buffer (pH 7.5) was incubated at 28 °C for 3 h. Another 100 μL reaction system containing 100 μM of **2** was incubated with 2.5 μM StvA2, 200 μM acetyl CoA, 5 μM StvP2, 100 μM Fdx, 5 μM FdR, and 3 mM NADPH in 50 mM Tris-HCl buffer (pH 7.5) at 28 °C for 3 h. The controls were conducted at the same conditions without the protein StvP2 or StvA2. All the reaction mixtures were then quenched with identical volume of methanol and centrifuged at the speed of $13,800 \times g$ for 5 min. The resulting supernatants were finally analyzed by HPLC-DAD.

**Sequential enzymatic conversion of 7 by StvM1, StvA2, and StvP2.** A series of biochemical conversions of **7** were performed in a 100 μL reaction system containing 20 μM StvM1 and/or 2.5 μM StvA2 or/and 5 μM StvP2 with 100 μM **7**, 4 mM NADPH, 200 μM acetyl CoA, 0.25 mM SAM in 50 mM Tris-HCl buffer (pH 7.5) in the presence or absence of 100 μM Fdx and 5 μM FdR at 28 °C for 3 h. All the reactions were terminated by identical volume of methanol, followed by centrifugation at $13,800 \times g$ for 5 min. Then the resulting supernatants were analyzed by LC–ESI–HRMS.

**Enzyme kinetics measurement of StvP2.** The gradient concentration (10, 20, 40, 80, 160, and 320 μM) of **3** were analyzed by HPLC-DAD. The peak area at $\lambda_{max}$ 432 nm was plotted as a function of the concentration of **3** using the Origin pro 2017C software.

All the substrates (**2**, **4**, and **6**) were dissolved in 50% aqueous methanol as stocking solution (16 mM). A series of descending concentration solutions (8, 4, 2, 1, 0.5, and 0.25 mM) of each substrate were made with Milli-Q water. 10 μL of each dilution was added into the reaction mixtures (2.5 μM StvP2, 50 μM Fdx, 2.5 μM FdR, and 4 mM NADPH in 100 μL 50 mM Tris-HCl buffer (pH 7.5) to incubate at 28 °C for 4 min. Then the reaction mixture was immediately terminated with identical volume of methanol, and centrifuged at $13,800 \times g$ for 5 min to leave the supernatant for HPLC-DAD analysis. The initial velocity was expressed in the product formation according to the calibration curve. The kinetic curve of each substrate was fitted using the Origin pro 2017C software with modified Hill equation[24] as the fitting model.

**Crystallization of substrate-free and substrate-bound StvP2.** For crystallization, protein StvP2 in TBS (20 mM Tris, 300 mM NaCl, and pH 8.0) was concentrated to 10 mg mL$^{-1}$ after purification by gel filtration. Crystallization was screened by the Mosquito robot (TTP Lab Tech) with 96-well plates using commercial solution kits from Hampton Research at protein to buffer 2:1, 1:1, and 1:2 ratio. Best diffraction crystals were grown from 0.3 μL of protein StvP2 with 0.6 μL of the reservoir solution (pH 5.6), consisting of 0.17 M ammonium acetate, 0.085 M sodium citrate, 24% PEG 4000 (w/v, polyethylene glycol 4000) and 15% glycerol, using the sitting drop crystallization protocol[40]. The native StvP2 crystal was grown at 20 °C for 2 weeks to full size. Substrate-bound StvP2 crystal was obtained by soaking the native StvP2 crystal into 50 mM substrate **2** in above reservoir solution (pH 5.6) at 20 °C in 6 h.

**X-ray data collection and structure determination.** Both substrate-free and substrate-bound crystals were flash frozen in a stream of liquid nitrogen before data collection. The X-ray diffraction data were collected at SSRF Beamline BL19U1 and were subsequently indexed, integrated, and scaled by XDSGUI program suite[41]. The initial phases were determined by molecular replacement using the PhaserMR (Phaser crystallographic software)[42,43] in CCP4i with the CYP105AS1 mutant structure in complex with compactin (PDB ID code: 4OQR) as a search model.

Crystal structure of native StvP2 was then built using Buccaneer CCP4i[44] and inspected by Coot[45]. Refmac5 CCP4i[46] was used for refinement of the structure. The native substrate-free StvP2 structure was determined at a resolution of 1.35 Å, which belonged to C222$_1$ space group with the cell dimensions of $a = 60.1$ Å, $b = 145.7$ Å, $c = 90.4$ Å, $\alpha = 90°$, $\beta = 90°$, $\gamma = 90°$, and one molecule in one asymmetric unit.

The substrate **2**-bound StvP2 structure was solved using the native StvP2 crystal structure as the search model. After refinement, the electron density of $2F_O$-$F_C$ map for the substrate **2** became clear. The restraint file of substrate **2** was generated and optimized using PRODRG Server[47] and AceDRG[48]. The substrate **2** structure was built using LigandFit PHENIX[43]. Further refinement was carried out using Refine. Phenix[43]. The substrate **2**-bound StvP2 crystal structure was determined at a resolution of 2.3 Å, belonged to P12$_1$1 space group, with cell dimensions of $a = 61.6$ Å, $b = 87.2$ Å, $c = 78.7$ Å, $\alpha = 90°$, $\beta = 112.3°$, $\gamma = 90°$, and two molecules in one asymmetric unit.

Both structure representations were prepared with PyMOL L (DeLano Scientific). The geometries of the final structures were evaluated by PHENIX and the resulting coordinates and structure factors have been deposited in the Protein Data Bank (PDB ID: 6M4P [https://www.rcsb.org/structure/6M4P or https://doi.org/10.2210/pdb6M4P/pdb], 6M4Q [https://www.rcsb.org/structure/6M4Q or https://doi.org/10.2210/pdb6M4Q/pdb]).

**Reporting summary.** Further information on research design is available in the Nature Research Reporting Summary linked to this article.

## Data availability

The X-ray structures of StvP2 alone or in complex with its substrate were deposited in the Protein Data Bank under accession codes 6M4Q [https://www.rcsb.org/structure/6M4Q or https://doi.org/10.2210/pdb6M4Q/pdb] and 6M4P [https://www.rcsb.org/structure/6M4P or https://doi.org/10.2210/pdb6M4P/pdb]. All other data generated and analyzed in this study are available within the article and the Supplementary Information. Source data are provided with this paper.

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

## Acknowledgements

This work was supported by the National Key R&D Program of China (2018YFA0903200), the Open Funding Project from State Key Laboratory of Microbial Metabolism (MMLKF18-11), and the Young Elite Scientists Sponsorship Program by China Association for Science and Technology (2019QNRC001). We thank Prof. Xudong Qu at Wuhan University to provide us the plasmids used for expression of Fdx and FdR. We thank Prof. Chengpeng Fan, Prof. Changjiang Dong, and Dr. Haigang Song for structure refinement and validation. The diffraction data were collected at the Shanghai Synchrotron Radiation Facility (SSRF, P.R. China).

## Author contributions

G.S., C.H., Q.M., X.C., Z.L., and Y.L. carried out cloning and analysis of the gene cluster, and constructed mutants; G.S., C.H., and M.L. isolated compounds and determined structures; G.S., C.H., and Q.M. expressed and purified proteins; G.S. and C.H. performed enzymatic assays; Q.M. and Z.Z. achieved and solved the crystal structures; G.S.

wrote the paper; Z.Z., Q.M., and Z.D. discussed and revised the paper. Y.S. conceived the study, analyzed the data, and wrote the paper.

## Competing interests

The authors declare no competing interests.
