## [Peer Review File · Nature Communications]

REVIEWER COMMENTS

Reviewer #1 (Remarks to the Author):

The authors characterize the last three steps in the biosynthesis of the antibiotic Streptovaricin C (1). The substrate for the last step (2) was identified and structurally characterized. The last step is formation of a methylenedioxy bridge (MBD), catalyzed by the cytochrome P450 monooxygenase StvP2. The structure of 2 indicated that two preceding steps are methylation of the C6 hydroxyl and acetylation of the C4 hydroxyl, which are not necessarily ordered but are required for formation of the MBD by StvP2. P450s are known to catalyze oxidative O-demethylation reactions that are generally thought to proceed by hydroxylation of the methyl group to form an unstable hemiacetal that decomposes to a hydroxylated substrate and a molecule of formaldehyde. Plant P450s generally catalyze MBD formation between an O-methoxy group and a vicinal hydroxyl group on an aromatic ring. This could be initiated by hydroxylation of the methyl group followed by reaction with the vicinal hydroxy group to form the MBD and water. However, 2 and two other identified substrates 4 and 6 exhibit neighboring ketone groups rather than hydroxyl groups. The products of the StvP2 catalyzed reaction suggest that formation of the MBD is associated with keto-enol tautomerization that is likely to enable MBD formation. Additionally, the structure of the StvP2 was determined by X-ray crystallography in the absence of the substrate and after soaking the crystallized enzyme with 2 added to crystallization medium. These structures are the first to characterize a P450 monooxygenase that catalyzes MBD formation, and this enzyme is unique in the need for the substrate to undergo ketone-enol tautomerization to form the product. The tautomerization may be facilitated by proton transfer relay system in the protein. In general, this work is well documented, and the results provide strong support for the authors' conclusions. These are novel findings that are likely to be of interest to broad readership.

There is a significant concern regarding the refinement of the structure of the ligand 2 in the X-ray crystal structure for the ligand complex. The PDB Validation Report indicates a relatively high number of bond length and angle outliers for the refined ligand, and the geometry of the naphthalene ring is poor. This is also evident in Figure 4b, where two hydroxyl groups on the naphthalene ring are only 2.2Å apart creating a clash between the non-bond atoms. The hydrogen bond between a water and the hydroxyl group on C25 is too short at 2.4Å. This short bond length would be more consistent with a cation if the phenol is ionized. Similarly, the H-bond between His-92 and Asp-89 at 2.3 Å is too short. The PDB Validation Report also indicates many clashes are evident between the ligand and the protein. Moreover, the C10 ketone is rotated away from the C6-methoxy is not accessible for formation of the MBD. These disparities undermine conclusions about the structure of the substrate and interactions of the substrate with protein.

These distortions may reflect either a problem with weighting steric restraints versus reciprocal space or real space fitting using Coot. It is also important to verify that initial model of 2, exhibits reasonable bond lengths and angles and that the restraint files used are compatible with the correct structure and are properly weighted. The input models should be examined carefully, and the model should be refined again with restraints that are weighted to reduce unreasonable values for bonds and angles. This may also change the distances for the H-bonds and relations to the amino acid sidechains. The new coordinates should be used to update the PDB deposition.

Have the authors refined the complex with a model for the alternative tautomer? Is it possible that binding to the P450 favors tautomerization prior to hydroxylation? Perhaps, it fits the density better than the that for 2.

As decomposition of the carbinol competes with formation of the MDP bridge did the authors test whether O-demethylated 2 and/or formaldehyde are produced by the enzyme in addition to 1?

The authors indicate that the initial phases for structure determination were obtained by molecular replacement using the structure of a catalase PDB:4QR. This seems very unlikely because the

catalase structure is very different that of P450s including the structure shown for StvP2 in Figure 4a. If true, what was the log likelihood score for the solution and the initial R values? How different is the conformation of the structure of StvP2 from other structures for P450s that have macrocyclic substrates?

The Instructions for Authors indicate that a stereo figure of the complete protein fold should be included. This would be helpful particularly if the heme was also displayed for orientation. This figure could be added to the Supplemental Figures.

Please make sure that information about the reproducibility of experiments, the number of biological and technical replicates, and statistical analysis for all experiments is provided.

Minor concerns:

"Co-crystal structure" was used 5 times in the text. The use of "Co-crystal" is not appropriate when the substrate was not co-crystallized with protein, but rather, the structure of the complex was obtained by soaking the crystallized protein with substrate as described in the Methods section.

Abstract, 3rd line from the end: "This work provides"

Page 4, last sentence: This sentence should be reworded as two more sentences for better clarity.

Page 5, line 4: suggest "but unexpectedly, no peak disappeared and diminished visibly."

Legend Fig 3. Define A and C for chart A in the legend.

Supplemental Table 4 should have a legend that describes the formula for the modified Hill equation as well as definitions for the constants and variables used in the equation. V_{max} and V_i are reported as turnover numbers that are typically denoted as rate constant, ie. k_{cat} rather than V_{max} .

Page 9: P450s are characterized by 12 rather than 13 generally conserved helices (A-L) and may have additional helices that are typically designated by a prime symbol and the letter of the nearest conserved helix. Please indicate where helix M resides in the structure?

Page 10: typo - Ile390 is hydrogen "bonded"

Page 14: "To convince the function" suggested alternative "To interrogate the function...."

Page 14: "(Fig. 5a(ii)) and no apparent involvement in...."

Page 18: The meaning of the first two sentences of the bottom paragraph is not very clear.

Page 18: typo "humongous"

Page 22: "obtain the seed cultures"

Page 28: Please indicate the composition and pH of the protein buffer used for the crystallization of the protein.

Why are the References split into two parts?

Reviewer #2 (Remarks to the Author):

The manuscript submitted by Sun et al. reported the functional characterization of StvP2 of *Streptomyces spectabilis* in the methylenedioxy bridge formation in the streptovaricine biosynthesis. Whereas the role of StvP2 in the biosynthesis was suggested in previous report (Liu et al. 2017), authors continued the biochemical characterization of StvP2 using several putative substrates and related methyltransferase (StvM1) and acetyltransferase (StvA2) to clarify the biosynthetic pathway. Authors also reported the crystal structure of StvP2 to characterize the importance of amino acid residues in methylenedioxy bridge formation with site-directed mutagenesis. The experiments were carefully conducted and the manuscript was clearly described to understand the role of StvP2 in streptovaricine biosynthesis and also to deepen our understanding of the P450s in unique methylenedioxy bridge formation and its role in unique microbial secondary metabolites. Whereas the manuscript is almost satisfactorily described, some additional information and revisions would be useful to increase the importance of this manuscript.

1) Whereas the reaction mechanism of StvP2 was discussed based on 3D-structure and site-directed mutagenesis, the role of catalytic residues still needs clarification. Whereas authors mentioned Asp242-Ser243 as highly conserved "acid-alcohol" pair in the I helix, this such information should be discussed more in details. As mentioned in the text, these residues are commonly Glu-Thr pair and some P450s with novel functionality such as methylenedioxy bridge formation in plants have different modification (see Mizutani and Sato, 2011 Arch. Biochem. Biophys. 507; 194). More adequate discussions are required. Also proper citation for a distinctive catalytic triad (Asp89-His92-Arg72) should be given. Whereas site-directed mutagenesis showed the importance of these residues, these residues would be not conserved in other MDB forming enzymes, suggesting these residues are not catalytic but rather ligand binding residues. More characterization and discussion are needed for reaction mechanism based on the comparison with other P450s.

2) Since StvP2 role was predicted in previous manuscript reported by Liu et al (2017), this information should be mentioned properly in Introduction.

3) Preparation/origin of some materials and methods should be clearly mentioned; for example, StvP2 production and details of purification and identification of metabolites should be mentioned in the main text even briefly. When these information was given in supplementary files, the information should be cited in the main text properly. Also manufacture information should be given more in details.

4) Some abbreviations, which are not so common, should be spelled out at once.

5) In Figure 6, the legend should be checked again, especially for the labeling of color.

Reviewer #3 (Remarks to the Author):

Reviewer Report

Uncovering the P450-catalyzed Methylenedioxy Bridge Formation in Streptovaricins Biosynthesis

(NCOMMS-20-12205-T)

Recommendation

- Publish after major revisions

Comments to the authors and editors:

General assessment:

The manuscript titled "Uncovering the P450-catalyzed Methylenedioxy Bridge Formation in Streptovaricins Biosynthesis" presents mechanistic insights of the methylenedioxy bridge (MDB) formation in streptovaricins. Genetic and biochemical experiments alongside with X-ray crystal structures of the previously identified cytochrome P450 StvP2 (1.35 Å) alone and in complex with its substrate 6-methoxy-streptovaricin C (2.3 Å) clarified the nature of the enzymatic reaction. Further structure-based mutagenesis experiments revealed the mechanism of the MDB formation

to be an intramolecular nucleophilic substitution resulting from hydroxylation through the heme core and the keto-enol tautomerization via a crucial catalytic triad (Asp89-His92-Arg72). The experimental design and the conducted procedures alongside with the acquired data and the conclusions in this manuscript are of very good quality and involve many different methodologies of natural product research. The significance of this manuscript is the first mechanistic revelation of MDB formation in natural product biosynthesis, catalyzed by the cytochrome P450 StvP2. Therefore, the presented manuscript is in general suitable for Nature communications, since it provides an important disclosure for scientists involved in natural product research. However, the manuscript and the SI contain several grammar and spelling mistakes, and some sentences should be rephrased to improve the quality of the manuscript. These errors could have been easily avoided through careful proofreading of the manuscript. For that reason, all language and formal aspects connected to this manuscript should be revised before publication. The manuscript is nonetheless clearly written, it is possible to understand the major findings, and provide enough details to reproduce the experimental procedures. Besides the language, there are minor aspects that should be considered during revision. The manuscript can be accepted for publication, after implementation of the suggested changes as described below.

Minor inconsistencies:

P2, L8–03: “Natural products are important sources of drug candidates and play a highly significant role in the drug discovery and development³...”

\ Please add the latest updated version of the review mentioned in reference 3 (reference 3 might be removed)

(Newman, D. J. & Cragg, G. M. Natural Products as Sources of New Drugs over the Nearly Four Decades from 01/1981 to 09/2019. *Journal of natural products*; 10.1021/acs.jnatprod.9b01285 (2020).)

P2, L10–13: “In addition to the basic scaffolds constructed with various simple building blocks, the complexity of natural products arise mainly from post-decorating enzymes to functionalize the inert skeleton⁴”.

\ Please replace ...“to functionalize the inert skeleton” and write “to functionalize the biosynthetic skeleton”, since many natural product skeletons or backbones are not biological inert.

P5, L6–10: “To identify the authentic intermediate of StvP2, about 45 mg of 6-methoxy-streptovaricin C (2) as a most promising candidate was purified from Δ stvP2 fermentation and characterized by LC-ESI-HRMS (Supplementary Fig. 8) and NMR (Supplementary Figs. 9-14 and Supplementary Table 1)”

\ Intermediate 2 was not described in the previous publication or mentioned in this manuscript beforehand; in fact, you stated that you could not detect any decreasing peak in your in vitro enzymatic conversion of the crude extract from Δ stvP2 catalyzed by StvP2. So how have you been able to identify candidate mass 2? Please explain this in 1–2 sentences before referring to the isolation and structural elucidation of 2.

P7, Figure 2b: Please indicate the biosynthetic changes in the structures of 3–6 according the respective genetic disruption; this would help to see the corresponding differences in the chemical structure.

P8, L14–17: Taken together, intermediate 2 is the most favorable substrate for StvP2, which is also in accordance with the fact that streptovaricin C, as the direct product of 2 catalyzed by StvP2, is the major product in wild-type strain.

\ As mentioned above; could you detect/quantify via the in vitro enzymatic conversion of the crude extract from Δ stvP2 catalyzed by StvP2 any conversion of 2. If not, please provide an explanation or hypothesis for this finding.

P11, L18–21: Please replace “These biochemical evidences proved the catalytic mechanism proposed above” and write, “These biochemical evidences supported the catalytic mechanism proposed above”.

\ As you described in the subsequent sentence, the outcome of these biochemical assays, determine the essential amino acids in StvP2, but does not completely prove the proposed catalytic mechanism in Fig. 4C.

P12,P13, L6,7/L1: “According to the bioinformatics analysis, three methyltransferase encoding genes exist in the streptovaricin biosynthetic gene cluster”

\ Please write in the first part of the sentence; “According to previously conducted bioinformatics analysis,.....” and add at the end reference 6

P14: “Apparently, stvA1, which is 10.1 kb away from the cluster, could be ruled out due to its no similarity and abnormal short in length (390 bp) compared with known acyltransferase encoding gene (~ 1.2 kb).”

\ Please provide an appropriate reference for this statement.

P14–15: To explore the function of StvA2 in MDB biosynthesis, in vitro assays were conducted in the presence of acetyl CoA as donor of acetyl group. To our surprise, neither of intermediates 7 and 8 were catalyzed by recombinant StvA2 (Supplementary Fig. 6).....”

and

P15: “Based on it, we speculate that 7 and 8 isolated from stvA2 are very likely an oxidized form of the initial substrates, which might be the reason for the failure of above in vitro assay”.

\ The supplementary Fig. 6 displays SDS-PAGE analysis of recombination proteins StvM1, StvA2 and StvP2, whereas Supplementary Fig. 60 shows the “biosynthetic network of acetylation and MDB formation confirmed by in vitro assay”. Figure 6 shows only that 7 is not converted through recombinant StvA2; Please correct the cross-reference and include the missing LC-ESI-HRMS chromatogram, which displays that intermediate 8 is not catalytically converted by StvA2.

P15: “An acetyltransferase usually catalyzes an acetyl group provided by the acetyl CoA to link to a nucleophilic group, such as hydroxyl group, amino group, sulphydryl group, etc. through nucleophilic attack mechanism”

\ Please provide for this statement an appropriate reference and change “sulthydryl” to “sulfhydryl”.

P17, Fig.6a: Please adjust the colors in the chromatogram to the scheme in 6b; in particular, compound 7 should be changed to grey, and the grey peak in chromatogram ix has to be changed to green.

P17, Fig.6b: Please change “melonyl CoA” to “malonyl CoA” and “mehtylmalonyl CoA” to methylmalonyl CoA”

\ The figure itself is good; however for the biosynthesis of the prostreptovaricin biosynthetic building block, I personally would like to see the complete biosynthetic pathway such as proposed in your former publication 2017; with the results in this manuscript, you can complement the biosynthetic investigation.

P18, L3–5: “To our knowledge, the StvP2 co-crystal structure with substrate 2 elucidated in this study is the first P450 enzyme responsible for MDB formation”.

\ This statement is invalid; there are several cytochrome P450 enzymes catalyzing this reaction (you mentioned that previously (also see Suppl. Fig. 5); you mean it is the first mechanistic characterization of a cytochrome P450 enzyme catalyzing MDB formation. Please rephrase accordingly.

P18 L21–32/P19 L1–3: This section seems to be misplaced for the discussion; you describe here in detail the efforts and the procedure to optimize the conversion catalyzed by StvM1 and StvA1 by using the reductase pair Fdx and FdR from *Spinacia oleracea*. Please transfer most of this paragraph to the result section (The O-acetylation at C-4 of C-6 methylated intermediate is necessary for StvP2-catalyzed MDB formation) and combine the residual part with the previous

paragraph (P18, L13–20).

P19, L4–15: This section has to be transferred to the result part and should contain cross-references to the SI (Supplementary Figure 60).

P19, L6: Please change "... (green and red peak) ..." to "... (green and yellow peak) ..."

P19, L13–14: "This might be due to the difference of catalytic efficiency between StvA2 and StvP2. The intermediate 1 transformed from 2 by StvP2 was accumulated due to the lack of timely conversion of StvA2".

\ Since this still belongs to the result part and you propose a biosynthetic network of acetylation and MDB formation (Suppl. Fig. 60), it would be nice to have experimental support for the different catalytic efficiency of StvA2 and StvP2 of compound 2 (which was isolated in huge quantity (45 mg)). You already showed complete conversion of 2 after incubation with StvA2 or StvP2, but an additional time dependent conversion reaction to get an approximation of the different kinetics of both enzymes would perfectly complete the result part of this manuscript.

P19 Last paragraph of discussion: "Our elucidation of the network comprising acetylation and MDB formation revealed remarkable catalytic versatility of StvA2. It could serve as an ideal target for engineering new natural products".

\ Except, for those two sentences, I personally miss a short paragraph to conclude the manuscript and highlight very briefly the significance and the outlook of your work. This becomes even more important, since a significant amount of the original discussion has to be transferred to the result section.

Major inconsistencies:

The most significant language and style inconsistencies are presented below; however not all inconsistencies are mentioned. Please carefully revise and rephrase the manuscript before publication.

Language inconsistencies:

Title: Uncovering the P450-catalyzed Methylenedioxy Bridge Formation in Streptovaricins Biosynthesis

\ Please write "Uncovering the cytochrome P450-catalyzed Methylenedioxy Bridge Formation in Streptovaricins Biosynthesis"

P2, L1: Please change "Streptovaricin C is a naphthalenic ansamycin antibiotics....." to "Streptovaricin C is a naphthalenic ansamycin antibiotic....."

P3, L1: "... and inevitable concerns..... "change to "... and inevitable concern..... "

P3, L3: "... Staphylococcus aureus is a common pathogens..... "change to "..... Staphylococcus aureus is a common pathogen"

P3, L5: "It puts forwards a higher requirement for drug development..."

\ What do you mean with "it"? The increasing number of drug-resistant pathogens? Please specify to eliminate the ambiguity. Otherwise, rephrase the sentence, for example. "As a consequence to deal with the increasing number of drug-resistant pathogens, improved drug development strategies are urgently required."

P3, L23–25 " It laid the foundation for development of potent anti-MRSA drug through biosynthesis-based structure modification or chemically semi-synthesis".

\ Please specify "It". The sentence was adapted from the previous publication from the discussion section; I think in this manuscript you should state the outcome from this study slightly different to underline, that the development of potent anti-MRSA drugs through biosynthesis-based structure modification or chemical semi-synthesis approaches has not started yet. For example: "The biosynthetic investigation of streptovaricin C in this study might provide the foundation for future development of potent anti-MRSA drugs through biosynthesis-based structure modifications or chemical semi-synthesis."

P4, L1: Please correct "...wildly...." to ".....widely"

P4, L14–15: Please change "....., but also set a promising ground to hunt for more members..." to "..., but also set a promising foundation to find more members...."

P5, L2: Please change ".....recombinant StvP2 (Supplementary Fig. 6) were failed with no streptovaricins formed in our previous investigation" to ".....recombinant StvP2 (Supplementary Fig. 6 failed to form streptovaricins in our previous investigation"

P5, L11-12: Please change "..., which probably due to the electron density change of the chromophore after MDB formation." To "..., which is probably due to the electron density change of the chromophore after MDB formation."

P5, L12: Please change "The pure 2 were then" to "Pure 2 was then"

P5, L18-20: " In addition, based on the fact in above biochemical results that oxidations at C-24 and C-28 have existed in both substrate and product,...."

\ The content of this sentence is clear, but please rephrase this half sentence.

P6, L2: Please change "... without MDB moiety were achieved through multiple genes knock-out..." to "...without MDB moiety were achieved through multiple gene knock-outs..."

P10,L4: Please change "... The O atom of carbonyl in Ile390 is hydrogen boned to the hydroxyl of C-25 and C-27,...." to " The O atom of carbonyl in Ile390 is bond to the hydrogen of hydroxyl of C-25 and C-27,..."

P13,L1: Please change "Of which, StvM1 is identity with class I S-adenosyl-methionine(SAM)-dependent methyltransferase in Streptomyces sp. NRRL B-1347,..." to "Of these three methyltransferases, StvM1 shares high similarity with the class I S-adenosyl-methionine(SAM)-dependent methyltransferase in Streptomyces sp. NRRL B-1347....".

P14, L8-10: Please change

"Apparently, stvA1, which is 10.1 kb away from the cluster, could be ruled out due to its no similarity and abnormal short in length (390 bp) compared with known acyltransferase encoding gene (~ 1.2 kb)."

To:

"Apparently, stvA1, which is 10.1 kb away from the cluster, could be ruled out due to the lack of sequence similarity and the unusual short nucleotide length (390 bp) compared with known acyltransferase encoding gene (~ 1.2 kb)."

P14, L20: Please change

"These indicated that StvA2 should be responsible for more than one O-acetylation with broad substrates during streptovaricins biosynthesis

" to "

These indicated that StvA2 should be responsible for more than one O-acetylation with different substrates during streptovaricins biosynthesis"

Methods

In general: "E. coli" not "E.coli"

P23, NMR: "double doublet" not "double doulet"

Dear Reviewers,

We really appreciate your very useful comments on our manuscript entitled “Uncovering the P450-catalyzed Methylenedioxy Bridge Formation in Streptovaricins Biosynthesis” (No. NCOMMS-20-12205-T) for **Nature Communications**. We have made point-by-point responses to each comment and have revised the manuscript accordingly. Changes are highlighted in yellow in the revised manuscript as required.

With best regards,

Yuhui

Response to reviewer #1

1) Q: There is a significant concern regarding the refinement of the structure of the ligand **2** in the X-ray crystal structure for the ligand complex. The PDB Validation Report indicates a relatively high number of bond length and angle outliers for the refined ligand, and the geometry of the naphthalene ring is poor. This is also evident in Figure 4b, where two hydroxyl groups on the naphthalene ring are only 2.2 Å apart creating a clash between the non-bond atoms. The hydrogen bond between a water and the hydroxyl group on C25 is too short at 2.4 Å. This short bond length would be more consistent with a cation if the phenol is ionized. Similarly, the H-bond between His-92 and Asp-89 at 2.3 Å is too short. The PDB Validation Report also indicates many clashes are evident between the ligand and the protein. Moreover, the C10 ketone is rotated away from the C6-methoxy is not accessible for formation of the MBD. These disparities undermine conclusions about the structure of the substrate and interactions of the substrate with the protein.

These distortions may reflect either a problem with weighting steric restraints versus reciprocal space or real space fitting using Coot. It is also important to verify that initial model of **2**, exhibits reasonable bond lengths and angles and that the restraint files used are compatible with the correct structure and are properly weighted. The input models should be examined carefully, and the model should be refined again with restraints that are weighted to reduce unreasonable values for bonds and angles. This may also change the distances for the H-bonds and relations to the amino acid sidechains. The new coordinates should be used to update the PDB deposition.

A: We have regenerated the restraint files using Prodrug server and ACEDRG CCP4i2, and made further refinement of the data. The geometry of ligand **2** has improved a lot, the new X-ray structure validation report is attached here for your reviewing. PDB deposition has also been updated accordingly. Fig. 4a, b and Supplementary Table 5 have been updated with new data, and

Supplementary Fig. 43 has been added in the revised manuscript.

In Fig. 4b, the naphthalene ring has been 2.4 Å apart now. There is no hydrogen bond between a water and the hydroxyl group on C-25 in our data or figures. The hydrogen bond between a water and the hydroxyl group on C-1 is 2.4 Å. The hydrogen bond between a water and the hydroxyl group on C-21 is 2.6 Å. The H-bond between His-92 and Asp-89 is now 2.7 Å. The PDB Validation Report has been updated and attached after refinement to improve the ligand **2** geometry. The C-11 ketone rotating has been corrected after refinement using the optimized restraint file. The improved geometry indicates that C-11 ketone and C6-methoxy can be accessible for formation of the MBD and complete the reaction.

2) Q: Have the authors refined the complex with a model for the alternative tautomer? Is it possible that binding to the P450 favors tautomerization prior to hydroxylation? Perhaps, it fits the density better than that for **2**.

A: After optimizing the restraint file of ligand **2**, the geometry of ligand **2** has improved greatly. Then, we have tried the ligand **2** tautomer as you suggested. However, we did not observe obvious improvements. The structure of these two ligands are very close, so we cannot confidently determine which one fits better just from our crystal data. So, we still used **2** as ligand in the model.

3) Q: As decomposition of the carbinol competes with formation of the MDB bridge, did the authors test whether O-demethylated **2** and/or formaldehyde are produced by the enzyme in addition to **1**?

A: We have tested the O-demethylated **2** in the biochemical conversion of **2** catalyzed by StvP2, and we indeed detected a trace amount of O-demethylated **2** formed in this reaction. As for formaldehyde, it cannot be detected in this reaction since its molecular weight (m/z 30.02600) is out of the limit of detection of mass spectrum.

4) **Q:** The authors indicate that the initial phases for structure determination were obtained by molecular replacement using the structure of a catalase PDB: 4QOR. This seems very unlikely because the catalase structure is very different that of P450s including the structure shown for StvP2 in Figure 4a. If true, what was the log likelihood score for the solution and the initial R values? How different is the conformation of the structure of StvP2 from other structures for P450s that have macrocyclic substrates?

A: Thanks for the comments. We are sorry to have messed up the PDB code 4OQR with 4QOR for the initial model of Molecular Replacement. Actually, we used PDB: 4OQR (Structure of a CYP105AS1 mutant in complex with compactin) as initial Molecular Replacement model for native 1.35 Å data. We use PDB advance search to find the structure with highest similarity. 4OQR shared 46% sequence similarity with StvP2, which is also the highest entry. The initial MR model is generated using Phyre2 server. After initial Phaser-MR and Refmac5 refinement, the $R_{\text{factor}}/R_{\text{free}}$ is about 0.41/0.46 and electron density map is pretty good (Figure 1 here). SOLU SET RFZ=4.9 TFZ=3.5 PAK=11 LLG=18 LLG=113. The file named Revise phaser MR is attached for your reviewing about the Molecular Replacement. After further BUCCANEER building and Refmac5 refinement, we can finally achieve $R_{\text{factor}}/R_{\text{free}}=0.16/0.18$ of 1.35 Å native data at C222₁ space group.

Figure 1 Electron density map after initial MR and refinement.

StvP2 shared high similarities with other P450 enzymes. We selected the first three entries from PDB database of highest sequence similarity (varies

from 35%-46%). We have superimposed these three structures to StvP2-ligand **2** with RMSD 1.18 for 4OQR (Structure of a CYP105AS1 mutant in complex with compactin); 0.955 for 5IT1 (*Streptomyces peucetius* CYP105P2 complex with biphenyl compound) and 1.344 for 5YSW (Crystal Structure Analysis of Rif16 in complex with R-L), respectively (Figure 2 here). The overall structure fold showed high similarity (not shown here). The ligand binding regions were displayed here. StvP2-ligand **2** is shown in purple. Among them, the macrocyclic ligand of 5YSW (Rifamycin L) resembles ligand **2** most. However, due to different catalysis mechanisms, the reaction centers point to the reaction groups, so the binding pocket differs dramatically.

Figure 2 Superimposition of StvP2 with other cytochrome P450 enzymes. The overall structure fold show high similarity. Only ligand region is displayed here.

5) Q: The Instructions for Authors indicate that a stereo figure of the complete protein fold should be included. This would be helpful particularly if the heme was also displayed for orientation. This figure could be added to the Supplemental Figures.

A: Thanks for the suggestion. We have added the stereo figure of the complete protein fold with heme displayed as Supplementary Fig. 43.

6) Q: Please make sure that information about the reproducibility of experiments, the number of biological and technical replicates, and statistical analysis for all experiments is provided.

A: We have supplemented the number of biological and technical replicates, and statistical analysis for all experiments in proper places.

7) Q: “Co-crystal structure” was used 5 times in the text. The use of “Co-crystal” is not appropriate when the substrate was not co-crystallized with protein, but rather, the structure of the complex was obtained by soaking the crystallized protein with substrate as described in the Methods section.

A: We have amended the inappropriate term in the revised manuscript.

8) Q: Abstract, 3rd line from the end: “This work provides”

A: We have changed “The work provides” to “This work provides” as suggested in the abstract.

9) Q: Page 4, last sentence: This sentence should be reworded as two more sentences for better clarity.

A: As you suggested, we have rephrased the sentence to “However, the enzymatic conversion of protostreptovaricin I purified from Δ stvP2 by the recombinant StvP2 failed to form MDB-contained streptovaricins in our previous investigation⁶. To validate the hypothesis above, we expressed and purified the recombination 6 × His-tagged StvP2 (Supplementary Fig. 6) in *E. coli* BL21(DE3), and conducted the StvP2-catalyzed *in vitro* assay using the whole crude extract from stvP2 in-frame deletion mutant⁶ as substrate.” that showed from the last line of page 4 to line 6 of page 5 in the revised manuscript.

10) Q: Page 5, line 4: suggest “but unexpectedly, no peak disappeared and diminished visibly.”

A: We have changed “but against to our initial expectation no peak was found disappear or decreased visibly.” to “but unexpectedly, no peak disappeared and diminished visibly.” that showed in line 10-11 of page 5 in the revised manuscript.

11) Q: Legend Fig 3. Define A and C for chart A in the legend.

A: We have defined “A” and “C” in the Fig. 3 legend that showed in page 8 in the revised manuscript.

12) Q: Supplemental Table 4 should have a legend that describes the formula for the modified Hill equation as well as definitions for the constants and variables used in the equation. V_{max} and V_i are reported as turnover numbers that are typically denoted as rate constant, ie. k_{cat} rather than V_{max} .

A: We have added a note in the bottom of Supplemental Table 4 to describe the modified Hill equation and to define the constants and variables used in the equation. In this equation, the reaction velocities are represented with the concentration of products formed per min and per μM enzyme. So the obtained V_{max} is comparable to the k_{cat} value according to the references: [23] Licata, V. J. & Allewell, N. M. Is substrate inhibition a consequence of allostery in aspartate transcarbamylase. *Biophys. Chem.* **64**, 225-234 (1997) and [24] Vuong, T. V. et al. Xylo- and cello-oligosaccharide oxidation by gluco-oligosaccharide oxidase from *Sarocladium strictum* and variants with reduced substrate inhibition. *Biotechnol. Biofuels* **6**, 148-161 (2013).

13) Q: Page 9: P450s are characterized by 12 rather than 13 generally conserved helices (A-L) and may have additional helices that are typically designated by a prime symbol and the letter of the nearest conserved helix. Please indicate where helix M resides in the structure?

A: We have revised this description in line 10 of page 9. And we have designated the additional helix as K' helix and indicated the additional helix in the protein structure of Fig. 4a that showed in page 12 in the revised manuscript.

14) Q: Page 10: typo - Ile390 is hydrogen “bonded”.

A: We have revised accordingly.

15) Q: Page 14: “To convince the function” suggested alternative “To interrogate the function...”.

A: We have revised “To convince the function” to “To interrogate the function” that showed in line 5-6 of page 15 in the revised manuscript.

16) Q: Page 14: “(Fig. 5a(ii)) and no apparent involvement in...”.

A: We have changed “(Fig. 5a(ii)) appearing no direct involvement in...” to “(Fig. 5a(ii)) and no apparent involvement in...” that showed in line 12-13 of page 15 in the revised manuscript.

17) Q: Page 18: The meaning of the first two sentences of the bottom paragraph is not very clear.

A: For better understanding, we have rephrased the sentences to “During our tracing for the vital substrate of StvA2, the high instability of reduced intermediate (the reduced form of **7** and **8**) could not be obtained and only be isolated as the unnatural naphthoquinone intermediate (**7** and **8**) in oxidation state. These unavailable substrates became a big obstacle for further study. We cannot make a breakthrough until the reductive cassette was introduced.” that showed in line 18-23 of page 19 in the revised manuscript.

18) Q: Page 18: typo “humongous”.

A: We have changed the typo “humongous” to “homologous” that showed in line 8 of page 20 in the revised manuscript.

19) Q: Page 22: “obtain the seed cultures”.

A: We have changed “obtain the seeds culture” to “obtain the seed cultures” that showed in line 13 of page 21 in the revised manuscript.

20) Q: Page 28: Please indicate the composition and pH of the protein buffer used for the crystallization of the protein.

A: We have added these information as suggested.

21) Q: Why are the References split into two parts?

A: We have merged the References into one part as Nature Communications required.

Response to reviewer #2

1) Q: Whereas the reaction mechanism of StvP2 was discussed based on 3D-structure and site-directed mutagenesis, the role of catalytic residues still needs clarification. Whereas authors mentioned Asp242-Ser243 as highly conserved “acid-alcohol” pair in the I helix, this such information should be discussed more in details. As mentioned in the text, these residues are commonly Glu-Thr pair and some P450s with novel functionality such as methylenedioxy bridge formation in plants have different modification (see Mizutani and Sato, 2011 Arch. Biochem. Biophys. 507; 194). More adequate discussions are required. Also proper citation for a distinctive catalytic triad (Asp89-His92-Arg72) should be given. Whereas site-directed mutagenesis showed the importance of these residues, these residues would be not conserved in other MDB forming enzymes, suggesting these residues are not catalytic but rather ligand binding residues. More characterization and discussion are needed for reaction mechanism based on the comparison with other P450s.

A: As you suggested, we have further clarified the role of the catalytic residues of StvP2 as showed from line 20 of page 11 to line 7 of page 12. The conserved “acid-alcohol” pair in the I helix has also been discussed more in details that showed in line 9-12 from bottom of page 9, and the corresponding

reference (Mizutani and Sato, Arch. Biochem. Biophys. **507**: 194 (2011)) has been added as [26].

We have added two references for the distinctive catalytic triad “Asp-His-Arg” in page 10 as [27] Hendle, J. et al. Crystallographic and enzymic investigations on the role of Ser558, His610, and Asn614 in the catalytic mechanism of *Azotobacter vinelandii* dihydrolipoamide acetyltransferase (E2p). *Biochemistry* **34**, 4287-4298 (1995) and [28] Kubiak, R. J. et al. Involvement of the Arg-Asp-His catalytic triad in enzymatic cleavage of the phosphodiester bond. *Biochemistry* **40**, 5422-5432 (2001).

We have given more characteristic and discuss on the reaction mechanism compared with other P450 enzymes that showed in the second paragraph of page 12.

2) Q: Since StvP2 role was predicted in previous manuscript reported by Liu et al (2017), this information should be mentioned properly in Introduction.

A: Although the role of StvP2 was predicted in our previous publication (Liu, Y. et al. *ACS Chem. Biol.* **12**, 2589-2597 (2017)), actually no direct evidence achieved at that time. It provides a crucial hint for further elucidating the function of StvP2 for the MDB formation of streptovaricins in this study. So we think it might be more logical to be mentioned in the inception of studying StvP2.

3) Q: Preparation/origin of some materials and methods should be clearly mentioned; for example, StvP2 production and details of purification and identification of metabolites should be mentioned in the main text even briefly. When these information was given in supplementary files, the information should be cited in the main text properly. Also manufacture information should be given more in details.

A: We have added more detail information and cited in the main text properly in the revised manuscript and Supplementary Information.

4) **Q:** Some abbreviations, which are not so common, should be spelled out at once.

A: We have added the full name of the abbreviations such as LC-ESI-HRMS, HPLC-DAD, UV-Vis, NMR, DTT, ALA etc. when they first appeared as suggested.

5) **Q:** In Figure 6, the legend should be checked again, especially for the labeling of color.

A: Thank you for your careful reading. We have revised the color of the peaks in the chromatogram of Fig. 6a to make it matched with the scheme in Fig. 6b and legend.

Response to reviewer #3

1) **Q:** P2, L8–03: “Natural products are important sources of drug candidates and play a highly significant role in the drug discovery and development³...” Please add the latest updated version of the review mentioned in reference³ (reference 3 might be removed) (Newman, D. J. & Cragg, G. M. Natural Products as Sources of New Drugs over the Nearly Four Decades from 01/1981 to 09/2019. Journal of natural products; 10.1021/acs.jnatprod.9b01285 (2020).)

A: We have replaced the reference 3 with the latest updated version of the review as suggested.

2) **Q:** P3, L10–13: “In addition to the basic scaffolds constructed with various simple building blocks, the complexity of natural products arise mainly from post-decorating enzymes to functionalize the inert skeleton⁴”. Please replace “...to functionalize the inert skeleton” and write “to functionalize the biosynthetic skeleton”, since many natural product skeletons or backbones are

not biological inert.

A: We have changed “to functionalize the inert skeleton” to “to functionalize the biosynthetic skeleton” that showed in line 13-14 of page 3 in the revised manuscript.

3) Q: P5, L6–10: “To identify the authentic intermediate of StvP2, about 45 mg of 6-methoxy-streptovaricin C (**2**) as a most promising candidate was purified from Δ stvP2 fermentation and characterized by LC-ESI-HRMS (Supplementary Fig. 8) and NMR (Supplementary Figs. 9-14 and Supplementary Table 1)”. Intermediate **2** was not described in the previous publication or mentioned in this manuscript beforehand; in fact, you stated that you could not detect any decreasing peak in your in vitro enzymatic conversion of the crude extract from Δ stvP2 catalyzed by StvP2. So how have you been able to identify candidate mass 2? Please explain this in 1–2 sentences before referring to the isolation and structural elucidation of **2**.

A: We have added a sentence for explanation before referring to the isolation and structural elucidation of **2** that showed in line 11-13 of page 5 in the revised manuscript.

4) Q: P7, Figure 2b: Please indicate the biosynthetic changes in the structures of 3–6 according the respective genetic disruption; this would help to see the corresponding differences in the chemical structure.

A: We have indicated the structural differences of compounds **1-6** with different colour, and supplemented corresponding figure legend in Fig. 2b that showed in page 7 in the revised manuscript.

5) Q: P8, L14–17: Taken together, intermediate **2** is the most favorable substrate for StvP2, which is also in accordance with the fact that streptovaricin C, as the direct product of **2** catalyzed by StvP2, is the major product in wild-type strain.

As mentioned above; could you detect/quantify via the *in vitro* enzymatic conversion of the crude extract from Δ stvP2 catalyzed by StvP2 any conversion of **2**. If not, please provide an explanation or hypothesis for this finding.

A: At the beginning of the native substrate investigation of StvP2, we conducted a StvP2-catalyzed *in vitro* assay at 28°C for overnight using the whole crude extract from *stvP2* in-frame deletion mutant as substrate. As a result, a new peak corresponding to **1** was detected but unexpectedly no peak disappeared visibly and no quantification available. It is actually due to the instability of substrate **2** during overnight reaction. Thus the reaction time was reduced in 3 hours when subsequent enzymatic conversions were carried on to avoid the degradation of the substrates.

6) Q: P11, L18–21: Please replace “These biochemical evidences proved the catalytic mechanism proposed above” and write, “These biochemical evidences supported the catalytic mechanism proposed above”.

As you described in the subsequent sentence, the outcome of these biochemical assays, determine the essential amino acids in StvP2, but does not completely prove the proposed catalytic mechanism in Fig. 4C.

A: We have changed “These biochemical evidences proved the catalytic mechanism proposed above” to “These biochemical evidences supported the catalytic mechanism proposed above” that showed in line 5-7 of page 12 in the revised manuscript.

7) Q: P12, P13, L6,7/L1: “According to the bioinformatics analysis, three methyltransferase encoding genes exist in the streptovaricin biosynthetic gene cluster”

Please write in the first part of the sentence; “According to previously conducted bioinformatics analysis,…” and add at the end reference 6.

A: We have revised the sentence and added the reference that showed in line 22-23 of page 13 in the revised manuscript.

8) Q: P14: “Apparently, *stvA1*, which is 10.1 kb away from the cluster, could be ruled out due to its no similarity and abnormal short in length (390 bp) compared with known acyltransferase encoding gene (~ 1.2 kb).”

Please provide an appropriate reference for this statement.

A: We have added two references as follows in line 5 of page 15 of the revised manuscript for this statement:

[29] Xiong, Y., Wu, X. & Mahmud, T. A homologue of the *Mycobacterium tuberculosis* PapA5 Protein, Rif-Orf20, is an acetyltransferase involved in the biosynthesis of antitubercular drug rifamycin B by *Amycolatopsis mediterranei* S699. *ChemBioChem* **6**, 834-837 (2005).

[30] Qu, Y. et al. Solution of the multistep pathway for assembly of corynanthean, strychnos, iboga, and aspidosperma monoterpene indole alkaloids from 19E-geissoschizine. *Proc. Natl. Acad. Sci. USA* **115**, 3180–3185 (2018).

9) Q: P14–15: “To explore the function of *StvA2* in MDB biosynthesis, in vitro assays were conducted in the presence of acetyl CoA as donor of acetyl group. To our surprise, neither of intermediates 7 and 8 were catalyzed by recombinant *StvA2* (Supplementary Fig. 6).....” and P15: “Based on it, we speculate that 7 and 8 isolated from *stvA2* are very likely an oxidized form of the initial substrates, which might be the reason for the failure of above in vitro assay”. The supplementary Fig. 6 displays SDS-PAGE analysis of recombination proteins *StvM1*, *StvA2* and *StvP2*, whereas Supplementary Fig. 60 shows the “biosynthetic network of acetylation and MDB formation confirmed by in vitro assay”. Figure 6 shows only that 7 is not converted through recombinant *StvA2*; Please correct the cross-reference and include the missing LC-ESI-HRMS chromatogram, which displays that intermediate 8

is not catalytically converted by StvA2.

A: We have corrected cross-reference and supplemented the LC-ESI-HRMS results of intermediate **8** incubated with StvA2 as Supplementary Fig. 60 and corresponding method in Supplementary Information.

10) Q: P15: “An acetyltransferase usually catalyzes an acetyl group provided by the acetyl CoA to link to a nucleophilic group, such as hydroxyl group, amino group, sulthydryl group, etc. through nucleophilic attack mechanism” Please provide for this statement an appropriate reference and change “sulthydryl” to “sulfhydryl”.

A: We have changed the typo “sulthydryl” to “sulfhydryl” as suggested and appropriate references (31-35) have also been added in line 6-7 from bottom of page 15 in the revised manuscript.

11) Q: P17, Fig.6a: Please adjust the colors in the chromatogram to the scheme in 6b; in particular, compound 7 should be changed to grey, and the grey peak in chromatogram ix has to be changed to green.

A: We have adjusted the colors of the chromatogram in Fig. 6a to make them matched with that in Fig. 6b that showed in page 18 in the revised manuscript.

12) Q: P17, Fig.6b: Please change “melonyl CoA” to “malonyl CoA” and “mehtylmalonyl CoA” to methylmalonyl CoA”.

The figure itself is good; however for the biosynthesis of the prostreptovaricin biosynthetic building block, I personally would like to see the complete biosynthetic pathway such as proposed in your former publication 2017; with the results in this manuscript, you can complement the biosynthetic investigation.

A: We have changed “melonyl CoA” to “malonyl CoA” and “mehtylmalonyl CoA” to methylmalonyl CoA” in Fig. 6b. As for the complement of the complete

biosynthetic pathway in Fig. 6, we are afraid to have enough space in the limit page although it is reasonable and helpful.

13) Q: P18, L3–5: “To our knowledge, the StvP2 co-crystal structure with substrate 2 elucidated in this study is the first P450 enzyme responsible for MDB formation”.

This statement is invalid; there are several cytochrome P450 enzymes catalyzing this reaction (you mentioned that previously (also see Suppl. Fig. 5); you mean it is the first mechanistic characterization of a cytochrome P450 enzyme catalyzing MDB formation. Please rephrase accordingly.

A: We have rephrased the sentence to “To our knowledge, the StvP2 co-crystal structure with substrate 2 elucidated in this study is the first mechanistic characterization of a cytochrome P450 enzyme catalyzing MDB formation.” that showed in line 9-10 of page 19 in the revised manuscript.

14) Q: P18 L21–32/P19 L1–3: This section seems to be misplaced for the discussion; you describe here in detail the efforts and the procedure to optimize the conversion catalyzed by StvM1 and StvA1 by using the reductase pair Fdx and FdR from *Spinacia oleracea*. Please transfer most of this paragraph to the result section (The O-acetylation at C-4 of C-6 methylated intermediate is necessary for StvP2-catalyzed MDB formation) and combine the residual part with the previous paragraph (P18, L13–20).

A: We agree that the section of P18 L21–32/P19 L1–3 (see P19 L27–33/P20 L1–10 in the revised manuscript) seems also appropriate in the Results as our initial thought. However, considering the authentic reduced enzymes from wild-type strain for the reduction of naphthoquinone to naphthol remain undefined, and on the other hand the balance of paragraph length, it seems more suitable to keep this part content in the Discussion.

15) Q: P19, L4–15: This section has to be transferred to the result part and

should contain cross-references to the SI (Supplementary Figure 60).

A: We have transferred the previous section of line 4-18 of page 19 to line 6-21 of page 17 in the Result section of “The O-acetylation at C-4 of C-6 methylated intermediate is necessary for StvP2-catalyzed MDB formation” with cross-references to the SI.

16) Q: P19, L6: Please change “... (green and red peak) ...” to “... (green and yellow peak) ...”.

A: We have changed the color of the peaks in the chromatogram of Fig. 6a to make it matched with the scheme in Fig. 6b that showed in page 18.

17) Q: P19, L13–14: “This might be due to the difference of catalytic efficiency between StvA2 and StvP2. The intermediate **1** transformed from **2** by StvP2 was accumulated due to the lack of timely conversion of StvA2”. Since this still belongs to the result part and you propose a biosynthetic network of acetylation and MDB formation (Suppl. Fig. 60), it would be nice to have experimental support for the different catalytic efficiency of StvA2 and StvP2 of compound **2** (which was isolated in huge quantity (45 mg)). You already showed complete conversion of **2** after incubation with StvA2 or StvP2, but an additional time dependent conversion reaction to get an approximation of the different kinetics of both enzymes would perfectly complete the result part of this manuscript.

A: Thank you for your very useful suggestion. As you suggested, we have supplemented the time dependent conversions of **2** after incubation with StvA2 or StvP2, respectively. These results obviously showed that the catalytic efficiency of StvP2 is higher than that of StvA2, which supported our deduction in the text very well. Additionally, in order to better understand the accumulation of **1** in the biosynthetic pathway network, we have also conducted another time dependent conversions of StvA2 with available compound **1** with the same conditions in parallel. It also showed that the

catalytic efficiency of StvA2 transforming **1** is lower than that of StvP2. This further supported our deduction of “The intermediate **1** transformed from **2** by StvP2 was accumulated due to the lack of timely conversion of StvA2”. These time course results have been added to the Supplementary Information as Supplementary Fig. 63 and cited in line 17 of page 17.

By the way, we initially even thought to carry on another parallel conversion of **21** or 25-acetoxy-6-methoxy-streptovaricin **C** to **21** or 25-acetoxy-streptovaricin **C** by StvP2 to perfectly verify difference of catalytic efficiency between StvA2 and StvP2 in another side. But it is really hard to achieve it due to unavailability of the intermediate **21** or 25-acetoxy-6-methoxy-streptovaricin **C**.

18) Q: P19 Last paragraph of discussion: “Our elucidation of the network comprising acetylation and MDB formation revealed remarkable catalytic versatility of StvA2. It could serve as an ideal target for engineering new natural products”. Except for those two sentences, I personally miss a short paragraph to conclude the manuscript and highlight very briefly the significance and the outlook of your work. This becomes even more important, since a significant amount of the original discussion has to be transferred to the result section.

A: As you suggested, we have supplemented a paragraph to conclude the manuscript and highlight briefly the significance and the outlook of our work in the last paragraph of discussion in page 20.

19) Q: Language inconsistencies: Title: Uncovering the P450-catalyzed Methylenedioxy Bridge Formation in Streptovaricins Biosynthesis. Please write “Uncovering the Cytochrome P450-catalyzed Methylenedioxy Bridge Formation in Streptovaricins Biosynthesis”.

A: We have changed the title to “Uncovering the Cytochrome P450-catalyzed Methylenedioxy Bridge Formation in Streptovaricins Biosynthesis” as suggested.

20) Q: P2, L1: Please change “Streptovaricin C is a naphthalenic ansamycin antibiotics...” to “Streptovaricin C is a naphthalenic ansamycin antibiotic...”.

A: We have changed the sentence to “Streptovaricin C is a naphthalenic ansamycin antibiotic...” as suggested.

21) Q: P3, L1: “... and inevitable concerns...” change to “... and inevitable concern...”

A: We have changed the sentence to “... and inevitable concern...” as suggested.

22) Q: P3, L3: “... *Staphylococcus aureus* is a common pathogens...” change to “... *Staphylococcus aureus* is a common pathogen”.

A: We have changed the sentence to “... *Staphylococcus aureus* is a common pathogen” as suggested.

23) Q: P3, L5: “It puts forwards a higher requirement for drug development...”
What do you mean with “it”? The increasing number of drug-resistant pathogens? Please specify to eliminate the ambiguity. Otherwise, rephrase the sentence, for example. “As a consequence to deal with the increasing number of drug-resistant pathogens, improved drug development strategies are urgently required.”

A: We have replaced the sentence “It puts forward a higher requirement for drug development to deal with these increasing drug-resistant pathogens.” with “As a consequence to deal with the increasing number of drug-resistant pathogens, improved drug development strategies are urgently required.” as suggested.

24) Q: P3, L23–25 “It laid the foundation for development of potent anti-MRSA drug through biosynthesis-based structure modification or chemically semi-synthesis”. Please specify “It”. The sentence was adapted from the previous publication from the discussion section; I think in this manuscript you should state the outcome from this study slightly different to underline, that the development of potent anti-MRSA drugs through biosynthesis-based structure modification or chemical semi-synthesis approaches has not started yet. For example: “The biosynthetic investigation of streptovaricin C in this study might provide the foundation for future development of potent anti-MRSA drugs through biosynthesis-based structure modifications or chemical semi-synthesis.”

A: We have replaced the sentence “It laid the foundation for development of potent anti-MRSA drug through biosynthesis-based structure modification or chemically semi-synthesis.” with “The biosynthetic investigation of streptovaricin C in this study might provide the foundation for future development of potent anti-MRSA drugs through biosynthesis-based structure modifications or chemical semi-synthesis.” as suggested.

25) Q: P4, L1: Please correct “...wildly...” to “...widely ...”.

A: We have corrected the typo “wildly” to “widely” that showed in line 3 of page 4 in the revised manuscript.

26) Q: P4, L14–15: Please change “..., but also set a promising ground to hunt for more members...” to “..., but also set a promising foundation to find more members...”.

A: We have changed the sentence to “..., but also set a promising foundation to find more members...” that showed in line 17 of page 4 in the revised manuscript.

27) Q: P5, L2: Please change "...recombinant StvP2 (Supplementary Fig. 6) were failed with no streptovaricins formed in our previous investigation" to "...recombinant StvP2 (Supplementary Fig. 6) failed to form streptovaricins in our previous investigation".

A: We have changed the sentence to "...recombinant StvP2 (Supplementary Fig. 6 failed to form streptovaricins in our previous investigation" as suggested.

28) Q: P5, L11–12: Please change "..., which probably due to the electron density change of the chromophore after MDB formation." To "..., which is probably due to the electron density change of the chromophore after MDB formation."

A: We have changed the sentence to "..., which is probably due to the electron density change of the chromophore after MDB formation." that showed in line 18-20 of page 5 in the revised manuscript.

29) Q: P5, L12: Please change "The pure 2 were then ..." to "Pure 2 was then ...".

A: We have changed the sentence to "Pure 2 was then ..." that showed in line 10 from bottom of page 5 in the revised manuscript.

30) Q: P5, L18–20: "In addition, based on the fact in above biochemical results that oxidations at C-24 and C-28 have existed in both substrate and product,...". The content of this sentence is clear, but please rephrase this half sentence.

A: We have rephrased this sentence that showed in line 1-3 from bottom of page 5 in the revised manuscript.

31) Q: P6, L2: Please change "... without MDB moiety were achieved through multiple genes knock-out..." to "...without MDB moiety were achieved through multiple gene knock-outs...".

A: We have changed the sentence to "...without MDB moiety were achieved through multiple gene knock-outs..." that showed in line 6-7 from bottom of page 6 in the revised manuscript.

32) Q: P10, L4: Please change "... The O atom of carbonyl in Ile390 is hydrogen boned to the hydroxyl of C-25 and C-27,..." to "The O atom of carbonyl in Ile390 is bond to the hydrogen of hydroxyl of C-25 and C-27,...".

A: According to the Reviewer 1's suggestion, we have regenerated the restraint files using Prodrug server and ACEDRG CCP4i2, and made further refinement of the data of StvP2 in complex with ligand **2**. The geometry of ligand **2** has improved resulting in the minor change of interaction between StvP2 and ligand **2**. So the interaction of C25-OH and Ile390 has changed to the interaction of C23-OH with Ile390. Accordingly, we have changed the sentence to "The O atom of carbonyl in Ile390 is bond to the hydrogen of hydroxyl of C-23 and C-27,..." as suggested and other related sentences that showed in line 7-13 of page 10 in the revised manuscript.

33) Q: P13, L1: Please change "Of which, StvM1 is identity with class I S-adenosyl-methionine(SAM)-dependent methyltransferase in *Streptomyces* sp. NRRL B-1347,..." to "Of these three methyltransferases, StvM1 shares high similarity with the class I S-adenosyl-methionine(SAM)-dependent methyltransferase in *Streptomyces* sp. NRRL B-1347...".

A: We have changed the sentence to "Of these three methyltransferases, StvM1 shares high similarity with the class I S-adenosyl-methionine(SAM)-dependent methyltransferase in *Streptomyces* sp. NRRL B-1347..." that showed in line 13-15 from bottom of page 13 in the revised manuscript.

34) Q: P14, L8-10: Please change "Apparently, stvA1, which is 10.1 kb away from the cluster, could be ruled out due to its no similarity and abnormal short

in length (390 bp) compared with known acyltransferase encoding gene (~ 1.2 kb).” To: “Apparently, stvA1, which is 10.1 kb away from the cluster, could be ruled out due to the lack of sequence similarity and the unusual short nucleotide length (390 bp) compared with known acyltransferase encoding gene (~ 1.2 kb).”

A: We have changed the sentence to “Apparently, stvA1, which is 10.1 kb away from the cluster, could be ruled out due to the lack of sequence similarity and the unusual short nucleotide length (390 bp) compared with known acyltransferase encoding gene (~ 1.2 kb).” that showed in line 2-5 of page 15 in the revised manuscript.

35) Q: P14, L20: Please change “These indicated that StvA2 should be responsible for more than one O-acetylation with broad substrates during streptovaricins biosynthesis” to “These indicated that StvA2 should be responsible for more than one O-acetylation with different substrates during streptovaricins biosynthesis”

A: We have changed this sentence to “These indicated that StvA2 should be responsible for more than one O-acetylation with different substrates during streptovaricins biosynthesis” that showed in line 13-15 of page 15 in the revised manuscript.

36) Q: Methods. In general: “*E. coli*” not “*E.coli*”

A: We have changed “*E.coli*” to “*E. coli*” that showed in line 15 of page 21 in the revised manuscript.

37) Q: P23, NMR: “double doublet” not “double doulet”

A: We have changed “double doulet” to “double doublet” as suggested that showed in line 11 of page 22 in the revised manuscript.

REVIEWER COMMENTS

Reviewer #1 (Remarks to the Author):

The authors establish the last three steps in biosynthesis of the antibiotic Streptovaricin C. The last step is formation of a methylenedioxy bridge (MDB), catalyzed by the cytochrome P450 monooxygenase StvP2. Structures of the StvP2 was determined by X-ray crystallography in the absence of the substrate and after soaking the crystallized enzyme with the substrate. These structures are the first to characterize a P450 monooxygenase that catalyzes MDB formation, and this enzyme is unique in the need for the substrate to undergo ketone-enol tautomerization to form the product. The tautomerization may be facilitated by proton transfer relay system in the protein. In general, this work is well documented, and the results provide strong support for the authors' conclusions. These are novel findings that are likely to be of interest to a broad readership. The revised manuscript has incorporated new data that strengthens the authors conclusions, and revisions to the text have greatly improved clarity.

Reviewer #2 (Remarks to the Author):

The revised manuscript integrated the suggestions/recommendation raised by this reviewer. Whereas the citation for catalytic triad (Asp89-His92-Arg72) observed in the BC loop was appreciated, these residues would be not conserved in other P450s as mentioned before. Also, references cited did not reflect P450 reactions, especially methylenedioxy bridge (MDB) formation, rather reported other chemical reactions such as dihydrolipoamide acetyltransferase or phospholipase. Involvement of these residues in the keto-enol tautomerization for further MDB formation in streptovaricins biosynthesis was suggested by site-directed mutagenesis experiment, the importance of these residues in MDB reactions should be more carefully discussed with the sequence comparison with other P450s. Such discussion based on the sequence comparison is still lacking in the revision. In fact, reaction mechanism shown in Figure 4c would be not conclusive for general MDB reactions, and this reaction mechanism would be mentioned in Discussion section instead of Results section.

Reviewer #3 (Remarks to the Author):

The paper can now be published as the authors have addressed most points sufficiently.

Response to Reviewer #2:

Q: The revised manuscript integrated the suggestions/recommendation raised by this reviewer. Whereas the citation for catalytic triad (Asp89-His92-Arg72) observed in the BC loop was appreciated, these residues would be not conserved in other P450s as mentioned before. Also, references cited did not reflect P450 reactions, especially methylenedioxy bridge (MDB) formation, rather reported other chemical reactions such as dihydrolipoamide acetyltransferase or phospholipase. Involvement of these residues in the keto-enol tautomerization for further MDB formation in streptovaricins biosynthesis was suggested by site-directed mutagenesis experiment, the importance of these residues in MDB reactions should be more carefully discussed with the sequence comparison with other P450s. Such discussion based on the sequence comparison is still lacking in the revision. In fact, reaction mechanism shown in Figure 4c would be not conclusive for general MDB reactions, and this reaction mechanism would be mentioned in Discussion section instead of Results section.

A: The catalytic triad (Asp89-His92-Arg72) of StvP2 reported in our work is a very special case in cytochrome P450 enzyme, which has not been published before. After extensive screening of these literatures up to date, only two articles (references 27 and 28) have reported a catalytic triad consisting of the three amino acids "Asp, His, Arg". Exactly, they did not reflect P450 reaction, especially methylenedioxy bridge (MDB) formation, rather reported other chemical reactions such as dihydrolipoamide acetyltransferase or phospholipase, just as the reviewer mentioned. However, considering the differences between this rare catalytic triad and the reported ones, we think that the above references could further indicate the particularity of the catalytic triad of cytochrome P450 enzyme StvP2 in our work.

As reviewer suggested, we have added an amino acid sequence alignment of

StvP2 with other cytochrome P450 enzymes as Supplementary Fig. 45 and more related discussions.

The reaction mechanism shown in Fig. 4c is aimed at the MDB formation in streptovaricins, but not a general MDB biosynthetic mechanism in other natural products, such as in some lignans and alkaloids mentioned in the manuscript. Considering that the reaction mechanism was proposed according to the crystal structure of StvP2 in complex with the substrate **2** and further proved by site-directed mutagenesis experiments described in Result section, it seems more reasonable and logic to leave the reaction mechanism in Results section than in Discussion section.

REVIEWERS' COMMENTS:

Reviewer #2 (Remarks to the Author):

The revision clarified the uniqueness of StvP2 in MDB formation. StvP2 is surely novel enzyme and this manuscript revealed the secret of StvP2 reaction. No further revision is needed.
Congratulations !!!